# Evaluation of Sustainable Recycled Products to Increase the Production of Nutraceutical and Antibacterial Molecules in Basil Plants by a Combined Metabolomic Approach

**DOI:** 10.3390/plants12030513

**Published:** 2023-01-23

**Authors:** Mariavittoria Verrillo, Gunda Koellensperger, Marlene Puehringer, Vincenza Cozzolino, Riccardo Spaccini, Evelyn Rampler

**Affiliations:** 1Dipartimento di Agraria, Università di Napoli Federico II, Via Università 100, 80055 Portici, Italy; 2Centro Interdipartimentale di Ricerca per la Risonanza Magnetica Nucleare per l’Ambiente, l’Agroalimentare, ed i Nuovi Materiali (CERMANU), Università di Napoli Federico II, Via Università 100, 80055 Portici, Italy; 3Department of Analytical Chemistry, Faculty of Chemistry, University of Vienna, Waehringer Strasse 38, 1090 Vienna, Austria; 4Vienna Metabolomics Center (VIME), University of Vienna, Althanstrasse 14, 1090 Vienna, Austria

**Keywords:** recycling biomasses, metabolomics, phenolic compounds, antioxidant capacity, antibacterial properties

## Abstract

Background: An important goal of modern medicine is the development of products deriving from natural sources to improve environmental sustainability. In this study, humic substances (HS) and compost teas (CTs) extracted from artichoke (ART) and coffee grounds (COF) as recycled biomasses were employed on *Ocimum basilicum* plants to optimize the yield of specific metabolites with nutraceutical and antibacterial features by applying sustainable strategies. Methods: The molecular characteristics of compost derivates were elucidated by Nuclear Magnetic Resonance spectroscopy to investigate the structure–activity relationship between organic extracts and their bioactive potential. Additionally, combined untargeted and targeted metabolomics workflows were applied to plants treated with different concentrations of compost extracts. Results: The substances HS-ART and CT-COF improved both antioxidant activity (TEAC values between 39 and 55 μmol g^−1^) and the antimicrobial efficacy (MIC value between 3.7 and 1.3 μg mL^−1^) of basil metabolites. The metabolomic approach identified about 149 metabolites related to the applied treatments. Targeted metabolite quantification further highlighted the eliciting effect of HS-ART and CT-COF on the synthesis of aromatic amino acids and phenolic compounds for nutraceutical application. Conclusions: The combination of molecular characterization, biological assays, and an advanced metabolomic approach, provided innovative insight into the valorization of recycled biomass to increase the availability of natural compounds employed in the medical field.

## 1. Introduction

The World Health Organization has estimated that more than 80% of the population in developing countries depends primarily on herbal medicine for basic healthcare needs [1]. The main substances in such medicines pertain to the families of the so-called secondary metabolites. This all-inclusive definition stands for heterogeneous groups of low concentrated products, which, although not being essential for the plant growth, play a critical role in their adaptation to the environment. These groups of metabolites trigger targeted functionalities and are involved in the resistance or resilience response to biotic and abiotic stresses or to a quantifying interaction with the signaling molecule [2]. The large variety of secondary metabolites found in plants can be summarized into five main classes: tannins, terpenoids, alkaloids, flavonoids, and phenolic compounds, according to their bio-synthetical pathways [3]. Particularly, phenolic compounds are identified as biologically active molecules with a wide range of applications as antioxidants, antimicrobial, and antimicrobial additives [4]. The majority of known phenolic molecules are synthesized either through the shikimic acid or by the malonate/acetate synthetic routes [5]. Mediterranean aromatic plants include a large range of species that produce a diversity of valuable bioactive molecules suitable to pharmaceutical, nutraceutical, cosmetic, and agri-food applications [6]. Basil (*Ocimum basilicum* L.) is an aromatic annual plant of the Lamiaceae family acknowledged as a valuable source of phenolic acids, (rosmarinic, chicoric, caffeic, and caftaric) or flavonol (quercetin, kaempferol) displaying antioxidant and antibacterial properties [7].

The industrial exploitation of natural substances requires a preliminary extraction and refining of these compounds from plant tissues. The low concentration and the striking sensitivity to retrieving conditions of the secondary metabolites, drives the research work to develop novel methodologies aimed at eliciting the target synthesis and thus facilitate the subsequent isolation steps, minimizing the alterations, and improving the extraction yield [8]. The application of synthetic agrochemicals can quantitatively influence plant metabolic processes and trigger quality changes to the pharmacological properties of secondary metabolites [9]. The residues of pesticides or mycotoxins found in herbal medicines and phytotherapies alter the quality of products with therapeutic properties in the treatment of various diseases [10]. Therefore, an increasing focus is devoted to sustainable approaches to promote the synthesis of secondary metabolites in aromatic plants thereby preserving the quality of the final products [11]. In this context, the natural organic fraction obtained from recycled biomasses, such as humic substances (HS) and compost teas (CTs) have been identified as valuable potential abiotic effectors or biostimulants due to their ability to influence directly and indirectly the plant metabolism [12,13]. These molecules can act as bioactive compounds or bioeffectors in various physiological processes, including nutrient up-take, ion transport, enzyme activities, and stress resilience, by promoting appraisable effects on biochemical reactions of primary and secondary metabolism, connected to antibacterial or antioxidant activity and anti-inflammatory action [14,15,16]. An important requirement to optimize these sustainable strategies is the careful characterization of molecular properties of organic extracts from recycled biomasses combined with the elucidation of metabolic cycles, activated by plant biostimulants. Solid state Nuclear Magnetic Resonance is a powerful tool for the detailed investigation of natural organic materials and the elucidation of structure-activity relationships. This non-destructive technique allows a direct analysis of complex organic substrates without pretreatments and purification steps, thus providing the actual distribution of functional groups and limiting the occurrence of technical artifacts. 

The plant biochemical and physiological processes fostered by external influences may be accessed through the omics sciences, which involve integrated analytical approaches to study the molecular responses induced in plants and other organisms by specific treatments or environmental conditions [17]. The large qualitative variability and quantity range of available metabolites in the plant kingdom are promising for the discovery of new active pharmacological substances. However, these compound variabilities in plants hamper isolation, detection, and characterization processes of bioactive molecules [18]. Metabolomics focuses on the complete analysis of all metabolites present in a biological system of interest [19]. To attain this goal, metabolomics is based on rapidly evolving technologies often integrated in systems biology to study the interactions between the different functional levels of biological organisms [20,21,22]. In recent years, liquid chromatography (LC) and ultra-high performance liquid chromatography (UHPLC) coupled with high resolution mass spectrometry (HRMS) techniques, have become state-of-the-art methods for multi-omics studies due to their ability to identify a large range of low abundant molecules including secondary plant metabolites [23].

In this study, natural organic compounds from recycled agricultural biomasses were employed as biostimulants on basil plants to improve the synthesis of specific molecules with nutraceutical or medical applications. The advantage of this strategy is related to the use of recycling biomasses to increase the production of antioxidant and antimicrobial natural compounds. Conversely, possible drawbacks are related to the reproducibility of composting processes due to the seasonal variation of available agro-industrial wastes. To face this issue, the on-farm composting methodology may represent an easy-to-handle solution for the careful check of fresh biomasses and composting conditions [24,25]. Moreover, an associated goal was to strengthen a comprehensive understanding of the use of recycled biomasses as renewable sources of tailored bioactive molecules for the biosynthesis of defined metabolites in aromatic plants. The data on molecular characterization of compost extracts obtained by solid state Nuclear Magnetic Resonance were correlated with outputs on yield, antioxidant, and antimicrobial properties of plant metabolites. The combination between nontargeted and targeted metabolomic approaches allowed the qualitative and quantitative detection of selected metabolites involving the structure-activity relationship between molecular features of applied compost derivates and activated metabolic processes. The cutting-edge challenge is the potential definition of a viable protocol to correlate the characteristics of recycled natural organic components to the yield and efficacy of specific plant products with pharmaceutical applications. 

## 2. Results 

### 2.1. Molecular Characterization of Compost Extracts

The NMR spectra of humic substances and compost teas (Figure 1) showed a consistent release in both organic extracts of apolar aliphatic (0–45 ppm) and aromatic (110–160 ppm) components, for which the global amount varied from the 35 to 51% of the total area (Table 1). The first spectral region (0–45 ppm) includes the alkyl-C resonances, associated to the methylene segment (-CH_2_-) in aliphatic chains of various lipid compounds, such as fatty acids, plant waxes, and bio-polyesters [24]. In water dissolved fractions of CT-ART and CT-COF samples, the multiple peaks of alkyl-C interval may also indicate the presence of low molecular weight compounds such as branched acids and syderophors from plant and microbial origin [26]. The less intense shoulders within the 35–45 ppm range (Figure 1), are attributable to the tertiary (CH) and quaternary (C–R) carbons in assembled rings of sterol derivatives, as well as to CH and CH_2_ groups in α and β position of peptidic moieties and branched alkyl compounds [24,25]. The sharp signals centered around 56 ppm combine different organic compounds related to either the methoxyl substituents on the aromatic rings of guaiacyl and syringyl units in lignin components or the C-N bonds in amino acid moieties [25,27]. The peaks in the O-alkyl-C chemical shift range (60–110 ppm) are assigned to monomeric units of carbohydrates and polysaccharide chains, such as cellulose and hemicellulose of plant tissue. The signals at 61/62 ppm represent the carbon nucleus in position 6, followed by the intense coalescence around 73 ppm formed by the overlapping resonances of carbon 2, 3, and 5, in pyranoside structures, while the most de-shielded peak at 105/6 ppm derives from the di-O-alkyl anomeric carbon in linked glucose units [13,25]. The lower intensity or lack of C resonances at 82 ppm associated to C4 of hexose structures tied through β 1→4 bond (Figure 1), suggest the prevalence of pentose units of hemicellulose as well as the incorporation of oligosaccharides and simple carbohydrates [13,24,25,26]. Moreover, the high field shift of anomeric C 1 towards 101 ppm in CT samples (Figure 1), further support the preferential solubilization of dissolved low molecular weight sugars not involved in the glycosidic bonds [25]. The broad bands extended along the aryl-C interval (116–140 ppm) are the un-substituted and C-substituted phenyl units of aromatic components from plant origin, while the subsequent signals shown in the phenolic region (140–160 ppm) are indicative of O-bearing carbons in the aromatic ring of lignin derivatives and phenolic components. Finally, the peak at 174 ppm corresponds to carbonyl and carboxyl groups of aliphatic acids, amino acid moieties and uronic acids of hemicellulose, while the less intense band around 165 ppm in CT samples originate from inorganic carbonates. 

The integrated structural properties of HS and CTs may be depicted by the dimensionless parameters calculated from the combined relative abundance of identified functions (Table 1). Although all compost extracts revealed a noticeable abundance of carbohydrates and polysaccharides, ranging from 24.7 to 27.6%, the HB/HI values indicated a prevalent hydrophobic characteristic for a HS and CTs samples (Table 1). The A/OA and ARM parameters clearly highlighted a differential contribution of apolar alkyl-C and aryl-C groups in specific organic samples. Humic extracts from coffee wastes and both CT fractions showed a preferential accumulation of aliphatic compounds, while the humic substances from artichoke compost were characterized by the larger content of aromatic materials, which represented about 35% of the total carbon (Table 1). 

The calculation of the LigR ratios further the selective dissolution of lignin fragments and bioavailable phenolic materials in humic and water extracts of processed organic biomasses supported (Table 1). The shared lower values of LigR in all extracts (Table 1) underlined a close correlation of spectral intensities associated to methoxyl groups (45–60 ppm) and O-aryl-C molecules (140–160 ppm) [25,27,28]. The lowest LigR of CT-COF suggested a preferential accumulation of dissolved phenolic compounds, distinct from lignin-derived molecules, during composting processes of coffee ground wastes and their release in water extracts [25,29]. Conversely, the slight larger LigR ratios shown by HS-COF and CT-ART may be related to a contribution of nitrogenated molecules within the broad bands in the 40–55 ppm NMR region (Figure 1) that indicate the inclusion of peptidic and amino acid moieties [13].

### 2.2. Phenological Parameter of Basil Plants

The data of fresh biomass of *Ocimum basilicum* plants highlighted an overall positive influence of specific organic extracts and applied rates on basil development, compared to control samples (Figure 2). 

The larger significant increase in biomass yield was found for the intermediate rate of 50 mg L^−1^ of HS-ART, CT-COF, and HS-COF, which raised the plant fresh weight by +25%, +18%, and 8%, respectively (Figure 2). The HS-ART and CT-COF maintained a valuable performance even for the lower and higher doses, although with opposite trends. The plant treatment with HS-ART at 10 mg L^−1^ and 100 mg L^−1^ enhanced the biomass by 22% and 18% in respect to the control (Figure 2). Conversely, the plant response to CT-COF showed an increment of 15% and 20% at the corresponding lower and higher doses, respectively. For HS-COF, both lower and larger applied concentration showed less marked effects comparable with the lowest biomasses increase found for pot trials with CT-ART (Figure 2). 

### 2.3. Antioxidant Capacity of Basil Extracts

The application of composted derivates to basil plants induced an increase in antioxidant activity and total phenolic content with a dose-response effect (Figure 3a,b).

The DPPH assay of methanolic extracts from plants treated with HS-ART and CT-COF at the maximum concentration (100 mg L^−1^) displayed the most intense antioxidant capacity measured as TEAC values equal to 55 and 49 μmol 100 mg^−1^, respectively, followed by the plant treatments with 50 and 10 mg L^−1^. Conversely, although the application at larger concentrations of HS-COF and CT-ART showed an average slight increasing response in respect to control, the calculated TEAC equivalent did not differ significantly to each other at *p* = 0.05 statistical threshold (Figure 3a).

These results are in line with a previous study on the bioactive effect of humic derivates on the antioxidant activities of different species such as hot pepper (*Capsicum annuum* L.) or rice (*Oryza sativa*) [30,31].

The same trend has been observed for the total phenol content (TPC) determined as gallic acid by Folin–Ciocâlteu methodology. The larger amount of TPC were exhibited in the leaves of basil plants treated with HS-ART and CT-COF followed by HS-COF and CT-ART (Figure 3b). Each treatment produced the best phenolic yields at the higher applied amount of HS and CT solutions, thereby revealing a linear decreasing amount of gallic acid equivalent at lower tested concentrations (Figure 3b).

### 2.4. Antibacterial Activity of Basil Methanolic Extracts

Antimicrobial activity against Gram-positive bacterial strains was found in the preliminary DDK screening of methanolic extracts, obtained from plants treated by HS or CTs compared to the control (Table 2). The best performance against *S. aureus* was observed for HS-ART and CT-COF followed by HS-COF at the optimal concentration of 50 mg L^−1^, with inhibition zones equal to 20.1, 18.4, and 10.3 mm, respectively. Conversely, the basil extracts from CT-ART trials, turned out in the lowest effective antibacterial control (Table 2). Furthermore, even at the largest applied rate (100 mg L^−1^), the methanol extracts of basil plants treated HS-ART and CT-COF, exhibited the greater antibacterial efficacy against *L. monocytogenes* and *E. coli*, compared to other treatments, with inhibition diameter of 11.2, 10.2, and 10.1, 9.7 mm, for HS-ART and CT-COF samples, respectively (Table 2).

A similar trend was observed for the minimum inhibitory concentration assay (Table 3), which highlighted larger effectiveness for the metabolite mixtures isolated from plant treatments with HS-ART and CT-COF against *Staphylococcus aureus* and *Enterococcus faecalis*. In contrast, a lower susceptibility was found for Gram-negative bacterial strains *Escherichia coli*, *Listeria monocytogenes,* and *Salmonella typhi*, which generally showed larger MIC values (Table 3). In particular, methanolic extracts of plants treated with the average dose (50 mg L^−1^) of HS-ART had the best antibacterial performance (Table 3), with a small MIC value against *Staphylococcus aureus* and *Enterococcus faecalis* (1.2 and 1.4 µg mL^−1^, respectively) followed by *Listeria monocytogenes* (MIC value of 1.3 µg mL^−1^), and *Escherichia coli* (1.5 µg mL^−1^).

### 2.5. Untargeted and Targeted Metabolomic Approach

In order to investigate the biosynthesis of biochemical intermediates in basil plants treated with HS and CTs, the primary and secondary metabolites were analyzed by LC-HRMS using a combination of untargeted and targeted analytical approaches. Targeted metabolomics identifies a predefined group of metabolites for which standards are available to provide a sensitive and accurate detection for metabolite quantification [32,33]. On the other hand, untargeted metabolomics focuses on the simultaneous measurement of as many unknown metabolites as possible in a biological system, enabling a more exhaustive picture on the changes of the complete metabolic profile based on the identification of compounds affected by an external treatment [34,35]. 

Untargeted metabolomics enabled us to detect a panel of 149 metabolites in vegetable extracts (Appendix A). Briefly, the major identified compounds were carbohydrates (30%), organic acids (16%), lipids (17%), amino acids (9%), aromatics (11%), other nitrogenous compounds besides amino acids (12%), and other compounds (2%). The chromatogram of the metabolites extracted from the basil leaves of the representative control sample is shown in Appendix A. Overall, the humic substances from artichoke and compost tea from coffee grounds at different doses induced increasing yields for most compounds compared to the control, particularly for aromatic metabolites. Hence, in order to understand the amount and composition of the metabolites involved in the differentiation between the analyzed samples, the outputs of untargeted metabolomics analyzed by Principal Components Analysis (PCA) (Figure 4a,b). The PCA models revealed a good grouping of all treated plants with suitable reproducibility between replicates with high statistical significance (*p* < 0.05) for both principal components. The score- and loading-plots of PCA diagrams outlined the differences among samples and the involvement of specific variables in the group differentiation.

For both HS and CTs, the PCA of untargeted analysis allowed a noticeable separation between treatments as related to both sources and application rates (Figure 4a,b). The differentiation of plant samples along the two principal components (PC1 and PC2) was based on the similar distribution of metabolites with minor differences for HS and CT treatments (Figure 4a,b), thus indicating the involvement of the same metabolic path in the biostimulant activities of the applied organic extracts. The first main component (PC1) represented 33.55% and 29.3% of the total variance of HS and CT fractions, respectively. The presence of metabolites such as raffinose and proline may indicate the activation of metabolic adaptation to a stress signal [36]. Particularly, raffinose is synthesized in response to some biotic or abiotic stresses, although its accumulation in plant cells also suggests a carbon storage mechanism [37]. Proline is also commonly produced in a high concentration in response to a variety of abiotic stress, while its catabolism can provide an energy supply to drive plant growth when stress mitigation is applied [38]. Metabolomics profiles of basil leaves were further discriminated along the positive values of PC2 that represented 18.9% and 16.8% of total variance for HS and CT. The various samples were statistically partitioned according to the concentrations of glucose-6-phosphate, fructose-6-phosphate, aconitic acid, mannose, proline, glutamine, salicylic acid, abscisic acid, jasmonic acid, cinnamic acid, naringenin, resveratrol with minor differences in metabolite types among humic and water extracts (Figure 4 a,b). 

Moreover, targeted metabolomics analysis allowed the quantification of 55 metabolites including different organic compounds such as saccharides (mono- and di-saccharides), organic acids, amino acids, and nitrogenous bases (Table 4). In accordance with the results derived from untargeted metabolomics, the application of HS-ART and CT-COF induced increasing yields for most of the compounds compared to the control or other treatments without a clear dose-dependent effect (Table 4). To detect the effects of organic treatments on the leaf metabolome, the targeted data of polar plant extracts was evaluated by PCA. The PCA of plants treated with HS from artichoke or coffee grounds composted wastes at different doses explained 70.75% of the total variance, with PC1 and PC2 accounting for 60.77 and 9.98%, respectively (Figure 5a). Overall, 101 metabolites were quantified using external standards in a range from 0.1 to 10 µM. The limits of detection (LOD) ranged between 0.03 and 0.9 µM for 90% of the compounds (LODs of 3-phosphoglycerate, glutathione, NADP+, S-Adenosyl-homocysteine, citrate, iso-citrate, dAMP, and dCMP were higher than 0.9 µM). The lower limits of quantification (LLOQs) ranged between 1 and 5 µM accordingly.

In addition, compounds characterized by hydroxyalkyl groups, mainly derived from major carbohydrates, like glucose, glucose-6-phosphate, glucose-1-phosphate, and galactose, were prevalent in HS-ART and CT-COF plant extract compared to the control (Table 4). The application of HS-ART at the maximum dose (100 mg L^−1^) promoted the synthesis of aromatic amino acids with a concentration of 9.14, 53.99, 17.32 nmol g^−1^ for phenylalanine, tryptophan, and tyrosine, respectively. Additionally, the treatment of basil with 100 mg L^−1^ CT-COF, encouraged the production of some phosphorylated sugars involved in plants’ essential biochemical processes such as photosynthesis or glycolysis. Regulated sugar phosphates included glucose-1-phosphate, glucose-6-phosphate, fructose-6-phosphate (3.04, 3.05, 3.81, nmol g^−1^), which were upregulated compared to the control (1.53, 1.55, 1.63 nmol g^−1^). The same trends have been observed for other organic acids such as citrate, fumarate, isocitrate, 3-phosphoglycerate, and other sugars involved in the biosynthesis of secondary metabolites, such as erythrose-4-phosphate and sedoheptulose-7-phosphate, and some secondary metabolites, such as rosmarinic acid and caffeic acid. Moreover, the heatmap derived from the targeted metabolomics data confirmed that the relative amount of the identified metabolites varied as a function of treatments, thus determining a different placement of plant samples in the score-plot (Figure 5b). Furthermore, the application of HS-ART showed an improvement in the yield of the following organic acids: citric, alpha-ketoglutaric, maleic, ribonic, oxalic, succinic, and lactic acid compared to the control (Figure 4 and Figure 5). In addition to the most abundant metabolites used in the PCA, the tested organic materials promoted the biosynthesis of intermediates represented by erythrose-4-phosphate and phosphoenolpyruvate (Figure 4 and Figure 5). These compounds act as intermediates of the primary carbon metabolism and are involved in the activation of shikimic or mevalonic acid biosynthesis of phenolic compounds or terpenes.

## 3. Discussion

### 3.1. Plant Development, Antioxidant and Antimicrobial Activities

#### 3.1.1. Plant Biomass

The acknowledged stimulative actions of HS or CTs on plant biochemical and physiological activity have been related to inducible effects on enzymatic pathways as hormone-like molecules [39,40]. The investigation of a “cause and effect” mechanism or structural-activities correlation between the organic extracts and plant growth have pointed out the concomitant influence of conformational properties and release of specific compounds on plant biochemical processes [25,41]. The nearness of hydrophobic and hydrophilic domains is a conducive feature for the self-aggregation of dissolved organic fraction into a micelle-like structure thereby favoring a surface tension activity with a potentially s adhesion capability on plant tissues [42,43]. These wrapping structures may act as an effective carrying-sink able to preserve and transport the active molecules, mainly represented by low molecular weight compounds including aromatic derivatives, lignin/phenol fragments, carbohydrates, and peptides [25,28,44]. The conformational changes of dissolved organic fractions upon dynamic interaction with the plant rhizosphere microenvironment drive the unfolding of organic layers and the release of bioavailable bioactive molecules in the proximity of root cell membranes [41,45]. In this work, the treatment of basil plants with water dissolved compost extracts revealed shared positive effects of the applied substances on plant development with an intensity modulation depending on specific chemical features. Both the HS and CT samples were characterized by defined hydrophobic components with suitable incorporation of hydrophilic O-alkyl compounds that may modulate pliable conformational flexibility and closer interaction with plant roots [27,46]. Besides the hydrophobic feature, the positive effects of HS-ART and CT-COF on basil fresh biomass may be related to the higher abundance of overall aromatic components, lignin units, and phenolic compounds elucidated by the NMR analyses (Table 1). For the humic extracts, the best performance of HS-ART can be explained by the higher content of bioavailable aromatic and lignin derivatives (Figure 1, Table 1). Similarly, for the application of CT_COF the greater release of dissolved phenolic compounds from the original source of coffee residues stressed by the LigR parameter indicates a preferential accumulation of small and active aromatic fragments, which may allow a prompt plant physiological response [47].

#### 3.1.2. Antioxidant Activity

Different natural substances in plant tissues act as antioxidant agents in biochemical processes, providing a significant protection against various diseases related to oxidative stress induced by free radical species (RNS) and reactive oxygen species (ROS) [48]. Generally, antioxidant capacity is directly correlated with the content of phenolic compounds [28,29,49]. Therefore, the role of phenolic/quinonic components as electron donors/acceptor in plant extracts, have been extensively studied [50]. 

The phenolic compounds contained in *Ocimum basilicum* leaves can be used as powerful antioxidants and free radical scavengers in the prevention of lipid peroxidation and generation of ROS [51,52]. 

The level of antioxidant capacities and TPC recorded in basil leaves seemed determined by the chemical features of applied humic substances and compost teas, suggesting a potential structure-activity relationship between molecular composition and biological activity of organic materials [53]. Both HS-ART and CT-COF extracts, which have shown a larger scavenger activity of methanolic extracts, were defined by a significant distribution of overall aromatic molecules and specifically of lignin and phenolic compounds as stressed by NMR results (Figure 1, Table 1). The ability of HS to increase the radical scavenger activity of plant metabolites may be related to the effect of humic molecules on the biosynthesis of secondary metabolites and particularly on the biochemical framework of shikimic acid [28,46]. The results of DPPH assay are in line with acknowledged studies that explained the effects of humic substances on the phenylpropanoid derivates and on the promotion of partial oxidative phosphorylation in mitochondria, a process strongly connected with the biomass production, and, consequently, the content of secondary metabolites [28,54]. As previously discussed, the positive role of HS-ART and CT-COF in stimulating the secondary plant metabolisms relies on the combination of hydrophobic features (Figure 1, Table 1) and the inclusion of relatively small bioactive phenolic molecules, as well as polar fragments, such as carbohydrates and peptides [25,55].

#### 3.1.3. Antimicrobial Activity

The data on inhibition zones and MIC showed positive effects on antimicrobial activity of basil leaves extracts from HS-ART and CT-COF treatments (Table 2 and Table 3). The claimed structural features and the release of bioactive molecules influenced the secondary metabolism of basil plants thus promoting larger yields of phenolic components, which are involved in antimicrobial activity [28]. The less antibacterial efficacy of basil methanolic extracts against some Gram-negative bacterial strains, could be justified by the higher resistance of these microorganisms [56]. Generally, leaves of basil, either fresh or dried, have a large amount of antioxidant aromatic compounds. Therefore, basil extract could be used as a natural preservative to extend the shelf life of perishable foods according to the UK Food Standards Agency [57]. The antioxidant features and immune-boosting properties of basil have also been associated with observed protection against *Helicobacter pylori*, a bacterial strain involved in chronic gastric ulcers [58]. Different processes can affect the antimicrobial activity of aromatic plant extracts such as geographical origin, plant quality, method of extraction, and the application of organic fertilizers [59]. Several works have pointed out the steady efficiency of polyphenols as antimicrobial compounds against Gram-positive compared to Gram-negative bacteria, due to cell walls linked to a molecularly complex outer membrane, that hamper the penetration of antibiotics including natural phenolic compounds [56,60]. 

### 3.2. Combined Nontarget and Target Metabolomic Approaches

The major challenge of this work is the combination of complementary analytical strategies including NMR, biological and bioactivity assays, and targeted/untargeted metabolomics to investigate the effect of green compost derivatives to streamline the biosynthesis of specific secondary metabolites in basil plants. The applied plant metabolomics workflow endowed the elucidation of main changes promoted by abiotic effectors in metabolic composition of plant extracts. The adopted approaches were able to identify and quantify the metabolites based on the comparison to real standards reducing false positives, which may lead to misinterpretation of affected biological mechanisms [61]. Additionally, merging the data from different analytical platforms enabled a comprehensive investigation of the effect of HS or CTs as biostimulants for aromatic plants. The advance of this integrated procedure is the characterization of the whole composition of the plant metabolome followed by unequivocal metabolite identification and absolute quantification of the changes in the expression of metabolites induced as a response to the application of the treatment by compost derivates [44]. The presented green metabolomics strategy is an innovative workflow that may support the applications of natural organic derivates tailored to the synthesis of potential nutraceutical molecules such as narigenin and resveratrol (Figure 4) or other secondary metabolites derivates of the shikimic acid pathway (Figure 5 and Figure 6). 

#### 3.2.1. Effect of Organic Derivates on Plants Biochemical Pathways to Produce Metabolites by Nutraceutical Application

The naringenin is a flavonoid belonging to the class of flavanones that has a bioactive effect on human health as an antioxidant, free radical scavenger, anti-inflammatory, enhancer of carbohydrate metabolism, and modulator of the immune system [62]. Conversely, resveratrol, as a representative of the polyphenol stilbenoid group, exhibits antitumor activity and is considered a potential candidate for the prevention and treatment of several types of cancer. The results obtained in this work confirmed a clear effect of compost derivatives on plant metabolism according to previous studies which reported that shikimic-derived compounds (flavonoids, some alkaloids such as isoquinoline alkaloids, tocopherols, and phenols) are affected by humic acids [44]. Specific biochemical reactions that evolved from primary metabolism play a key role in the plant’s interaction with its environment and are strongly linked to the production of secondary metabolites [11]. Our metabolomics analyses have confirmed the occurrence of multiple interactions with metabolic pathways in accordance with the literature describing the effect of organic products on plant metabolism [45]. Green metabolomics based on the combination of targeted and non-targeted analysis further enlightens the relationship between chemical feature of compost extracts and the elicited yield of phenolic compounds, such as caffeic acid and rosmarinic acid. Rosmarinic acid displays important biological activities for memory enhancement such as circulation improvement, strengthening of vulnerable blood vessels, inflammation, and cerebral disorders [47,63]. 

Concomitantly, the metabolomic outputs match the data on plant development: in fact, a pronounced photosynthetic activity and protein synthesis fit with the high amino acid content found in plant treatments with organic bio-effectors. The accumulation of abscisic acid in basil plants treated with HS-ART or CT-COF is probably facilitated by glutamate decarboxylase. Additionally, the high content of aromatic amino acids such as tryptophan, tyrosine, and phenylalanine found in plants treated with HS-ART and CT-COF, suggests a higher production of these secondary metabolites as a response to the treatment (Table 1). 

#### 3.2.2. Effect of Compost Derivates on Plants Phenylpropanoid Derivatives

Notably, the significant presence of phenylalanine and tryptophan provides a substrate for the phenylpropanoid route [64], which is a critical step for the biosynthesis of many secondary plant products, such as anthocyanins, lignin, and phenols. The presence of chlorogenic and shikimic acids also confirms an enhanced synthesis of secondary metabolites (Figure 4, Figure 5 and Figure 6). Phenylalanine is a precursor for a variety of secondary plant products, such as anthocyanins, lignin, and phenols. On the other hand, tryptophan is the preliminary intermediate of indolacetic acid, which is involved in cell expansion and many other regulatory processes. Synthesis occurs through the shikimate reactive chain, followed by the metabolic pathway of branched aromatic amino acid [64]. Our results on the use of compost extracts corroborate previous hypotheses [54,55] about the induction of HS on the expression of phenylalanine ammonia lyase, which catalyzes the first phase in phenylpropanoid biosynthesis, transforming tyrosine to *p*-cumaric acid and phenylalanine to *trans*-cinnamic acid. The stimulation of these enzymes was associated to an uptake of exogenous phenolic compounds in plant leaves. Furthermore, the increase in the relative content of sugars such as glucose and its two phosphorylated forms as well as fructose and galactose are reflected by an increase in photosynthetic efficiency. Glucose-1-phosphate is a key intermediate in several major carbon anabolic fluxes, such as sucrose, starch, and cellulose biosynthesis while glucose-6-phasphate shows an effect on the oxidative pentose phosphate cycle [46]. With respect to the hypothesized structural-activity correlation, the uneven response found in the biological and antimicrobial assays in respect to the different concentrations of natural organic materials is underpinned by the physical-chemical behavior of compost extracts when applied as dissolved water fractions. As previously noticed, the contiguous hydrophobic and hydrophilic domains may form in solution flexible pseudo-micelles, which act as protective carriers of included apolar and polar bioactive molecular fragments. The dynamic equilibrium of dissolved HS and CT components is iteratively regulated by the environmental condition (pH, soil moisture, ionic strength, root exudates, active mineral surfaces, etc.), thus acting as a driving parameter for the mobility, molecular conformation, and functional group distribution, which determine the HS/CT-mediated bio-stimulant activity [27,41]. In the rhizosphere environments, the large surface tension triggered by the hydrophobic components fosters the interaction with the solid root systems [25,45]. However, depending on the variable condition of the rhizoplane, the organic assembly may undergo unfavorable rearrangement and overcome the Critical Micelle Concentration threshold with a consequent alteration of aqueous solubility and diffusivity properties [42,65] that may bias the hypothesized structural activity mechanism and affecting the dose-response effects. Compared to the plant response to applied doses, increasing effects were shown by the addition of 50 mg L^−1^ rates of both HS and CT to the soil, which was identified as the optimal dose. Higher concentrations (100 mg L^−1^) possibly promoted a closer aggregation of aromatic and alkyl hydrophobic clusters or integer ligno-cellulose structures with a small chance of structural modification and irregular release of bioactive molecules [25]. In this context, an alternative process on plant roots was related to the activation of physiological responses based on reduced root hydraulic conductivity and induced stress signals [45,53]. Conversely, the average dissolved amount (50 mg L^−1^) of compost extracts may have undergone the above-sketched structural rearrangement with a suitable interaction with plant rhizosphere and unfolding of bioactive effects depending on the specific molecular components. The uneven stimulating effects observed for the lower concentration (10 mg L^−1^) suggested the occurrence of a limiting interference of the two mechanisms. A comprehensive understanding of the mechanisms that underpin the stimulation of alternative metabolic biosynthetic reactions may provide useful information to attempt a tailored application of specific organic materials on selected crops. 

## 4. Materials and Methods

### 4.1. Humic Substances (HS) and Compost Teas (CTs)

The composts were obtained from the aerobic processing of both recycled chipped artichoke residues and coffee grounds wastes, carried out in the composting facility of the Experimental farm of University of Naples Federico II [28]. The mature composts were air dried, sieved at 2 mm to remove undecomposed plant debris and used to isolate bioactive fractions, namely humic substances (HS) and Compost teas (CT). For humic substances, an aliquot of 100 g of each compost sample was suspended in 500 mL 0.1 mol L^−1^ KOH and mechanically shaken for 24 h. The suspension was then centrifuged at 7000 rpm for 20 min and filtered with glass fiber. The recovered pellets were resuspended in water, brought to pH 7, and dialyzed against deionized water (1-kD cutoff spectrapore membrane) until an electrical conductivity < 0.5 dS m^−1^ and finally freeze-dried. The isolation of CTs was performed with 200 g of each compost weighted into a gauze bag and subsequently placed in a plastic beaker containing 1 L of distilled water (*w*/*v* 1/5). The extraction of dissolved components is a static process based on air insufflation (5 min every 3 h) with an automatic aeration pump device. After seven days, the CT solution was freeze-dried and stored a 4 °C.

### 4.2. Experimental Design, Plant Growth, Sampling, and Analyses

The pot trials for the cultivation of basil plants were prepared with the surface layer (0–15 cm) of agricultural soil classified as a Vertic Xerofluvent. The soil had a clay loam texture (44.6%, 28%, and 27.4%, for sand, silt, and clay, respectively), alkaline pH (8.6) 1.11 g kg^−1^ total N, 10.5 g kg^−1^ organic carbon, 11 mg kg^−1^ of NaHCO_3_-extractable P. 

Basil plants were sowed in the substrate made by soil sieved to 5 mm and thoroughly mixed with quartz sand at the ratio of 2:1 (*w*/*w*) to obtain 1 kg of final mass for each pot. The nutrients supply for plant growth was composed by ammonium nitrate (NH_4_NO_3_) 150 mg N kg^−1^, triple superphosphate (TSP) f 75 mg P kg^−1^ and potassium sulphate (K_2_SO_4_) 160 mg P kg^−1^. Following the addition of mineral fertilizers both HS and CTs from composted biomasses were tested as water suspension at three different concentrations set at 10, 50, and 100 mg L^−1^ applied once a week for 4 weeks, corresponding to a final amount of 2.0, 10.0, and 20.0 mg Kg^−1^ soil, respectively.

The pot experiment was conducted from March to June 2018, under greenhouse conditions (25–33 °C, daily temperature range) with the following design:
A: CTRL: Control (H_2_O) + mineral fertilizers; B: A + HS artichoke 10 mg L^−1^; C: A + HS artichoke 50 mg L^−1^; D: A + HS artichoke 100 mg L^−1^;E: A + CT artichoke 10 mg L^−1^; F: A + CT artichoke 50 mg L^−1^; G: A + CT artichoke 100 mg L^−1^;H: HS coffee 10 mg L^−1^; I: A + HS coffee 50 mg L^−1^; L: A + HS coffee 100 mg L^−1^;M: CT coffee 10 mg L^−1^; N: A + CT coffee 50 mg L^−1^; O: A + CT coffee 100 mg L^−1^.

Each treatment was replicated four times for a total of 52 pots. Harvested leaves were weighted, frozen in liquid nitrogen, and stored at −80 °C. 

### 4.3. Extraction of Plant Leaf Metabolites

For the selective isolation of target metabolites about 60 ± 0.5 mg of pooled homogenized plant material was weighed into pre-chilled 2 mL Eppendorf tubes and added with 1 mL of water/methanol/chloroform mixture (1:3:1 ratio) pre-cooled at −20 °C. The samples were mixed for 60 s and incubated for 30 min at 70 °C to inhibit the activity of possible co-extracted enzymes. The mixtures were then centrifuged for 10 min at 10,000 rpm and 4 °C; the supernatants were quantitatively transferred into 2 mL Eppendorf tubes and combined with Milli-Q water (400 μL) to facilitate the separation of polar and apolar phases. All extracts were finally stirred for 30 s and centrifuged for 10 min at 4 °C at 10,000 rpm. A volume of 400 μL was collected from the upper phase, transferred into 1.5 mL glass tubes for LC–MS analyses, dried using speed-vacuum and stored at −80 °C [66,67]. 

### 4.4. ^13^C-CPMAS-NMR Spectroscopy 

The NMR spectra of powdered homogenized samples of humic substances and compost teas were on Bruker AV-300 instrument equipped with 4 mm wide-bore MAS probe. For the identification of molecular characteristics, the ^13^C-CPMAS-NMR spectra were split into six chemical shift regions: alkyl-C (0–45 ppm); methoxyl-C and N-alkyl-C (45–60 ppm); O-alkyl-C (60–110 ppm); unsubstituted and alkyl-substituted aromatic-C (110–145 ppm); O-substituted aromatic-C (145–160 ppm); carboxyl- and carbonyl-C (160–200 ppm). The structural features of organic materials may be explained and synthesized by dimensionless structural indexes derived from the relative amount of NMR functional groups [13,25]:the hydrophobic index is the ratio of signal intensities of apolar alkyl and aromatic C components over those of hydrophilic C moleculesHB/HI = Σ[(0–45) + (45–60)/2 + (110–160)]/Σ[(45–60)/2 + (60–110) + (160–190)];the aromaticity index compares the number of aromatic compounds to that of the alkyl groupsARM = [(110–160)/Σ(0–45) + (60–110)];the alkyl ratio is determined by the relative abundance of apolar aliphatic molecules over that of the polar-alkyl fractionsA/OA = (0–45)/(60–110);the lignin ratio, relates the area of methoxyl-C and N-alkyl groups to that of O-aryl-C functionsLigR = (45–60)/(145–160).

The HB, ARM and A/OA parameters are widely applied to estimate both the biochemical stability of natural organic matter, as well as the influence of structural properties on the biostimulant activities of organic extracts [24,26]. The lignin ratio (LigR) may be used to evaluate the relative contribution of lignin-components and that of C-N carrying moieties as well as to discriminate between signals owing to lignin and those characteristic to other phenolic compounds [13,25].

### 4.5. Antioxidant Activity of Methanolic Basil Extracts 

#### 4.5.1. Sample Extraction

To isolate the plant metabolites, 10 mg of dried samples of basil leaves from each treatment replicate were ground in a mortar and extracted with 5 mL of acidified methanol (1% HCl) and placed at room temperature for 1 h [68]. The mixture was then centrifuged at 7500 rpm for 10 min transferring the supernatant d into a 10 mL tube to determine the total phenolic content and DPPH radical scavenging.

#### 4.5.2. Total Phenolic Compounds (TPC)

The TPC were determined by the Folin–Ciocâlteu colorimetric reference method [66]. A precise aliquot (200 µL) of leaves extracts was oxidized with 200 µL of 1 N Folin–Ciocâlteu reagent, and subsequently neutralized with Sodium carbonate (7.5%). After incubation for 1 h at 25 °C, the absorbance of the mix was measured at 730 nm with a UV-VIS spectrophotometer. The TPC was estimated through external calibration curve and expressed as milligrams of gallic acid equivalent (mg of GAE) per gram of dry weight (DW). 

#### 4.5.3. DPPH Radical Scavenging Assay

To analyze the free radical scavenging activity of holy basil, 0.1 mL of methanolic basil extracts were incubated in 0.4 mL of 0.1 M Tris–HCl (pH 7.5 ± 0.1) and 0.5 mL of 0.3 mM DPPH in the dark at room temperature for 20 min. The absorbance of the final solution (*A*_sample_) was measured at 517 nm and compared to the absorbance of the reference (*A*_control_) consisting in 80% methanol, 0.1 M Tris–HCl, and 0.5 mL of 0.3 mM DPPH. The free-radical scavenging was calculated according to the following equation: % DPPH scavenging = (1 − *A*_sample_/*A*_control_) × 100. 

The % DPPH free-radical scavenging was then compared to a trolox standard curve, and antioxidant capacities were expressed as trolox equivalent antioxidant capacities (TEAC, mmol of trolox equivalents/100 g DW).

### 4.6. Antimicrobial Activity of Leaf Methanolic Extracts

The antibacterial activity of basil extracts was evaluated by diffusion disk assay (DDK) and the broth microdilution (MIC) method [29]. The tested bacterial strains were kept at −80 °C in Luria Bertani (LB) broth with 20% glycerol and activated for the analysis by incubation on agar substrate at 37 °C for 18 h. Bacterial strains used in this work include: *Staphylococcus aureus ATCC 6538P*, *Enterococcus faecalis ATCC 29212*, *Listeria monocytogenes* (clinical isolates), *Salmonella typhi ATCC14028*, *Pseudomonas aeruginosa ATCC 27355* and *Escherichia coli ATCC 35218.* Three independent experiments were performed for each DDK and MIC value. The DDK was performed according to the National Committee for Clinical Laboratory Standards (NCCLS) standard method, using 25 μL of 12.5, 25, 50, 100, and 200 mg/mL of each plant’s extracts which correspond to 0.3125, 0.625, 1.25, 2.5, and 5.0 mg/disk, tested dilutions, respectively. The inoculum of the colonies was standardized by replacing the nutrient agar with sterile saline solution up to 10^8^ CFU mL^−1^ (0.5 McFarland), which is equivalent to 50% transmittance at 580 nm (Coleman model 6120, Maywood, IL, USA). Subsequently, 200 μL of the bulk suspensions was placed onto the surface of Mueller–Hinton agar. Disks (6.0 mm diameter) were impregnated with 25 μL of each sample, placed on the agar Petri dish and incubated at 37 °C for 24 h. Sterile distilled water (20 μL), bovine serum albumin (BSA) and ampicillin (30 μg) were used control negative and positive reference, respectively. The total diameters were measured by considering the size of the inhibition zones. Each experiment was performed in triplicate. 

The second antimicrobial assay was performed by the broth microdilution method in a Mueller–Hinton broth medium by using sterile 96-well polypropylene microtiter plates. The microbial inoculum size used was 1 × 10^6^ cells mL^−1^ (NCCLS, 1993). Twofold serial dilutions of different samples were carried out to obtain concentrations ranging from 10 to 1000 μg mL^−1^. Then, the bacterial cells were inoculated from an overnight culture at a final concentration of about 5 × 10^5^ CFU mL^−1^ per well and incubated with different samples overnight at 37 °C. The minimal inhibitory concentration (MIC) values—that is, the lowest concentration of material that inhibited the growth of microorganisms after 24 h of incubation at 37 °C—were estimated by measuring the spectrophotometric absorbance at 570 nm. 

### 4.7. Metabolomic Approaches

#### 4.7.1. Standard References 

The metabolite standards were purchased from Sigma, Carbosynth (Compton Berkshire, UK), Avanti Polar Lipids, Inc. (Alabaster, AL, USA), Fluka and Merck (Vienna, Austria) and used to prepare a multi-metabolite mix (309 metabolites) (a comprehensive list of the standards can be found in Appendix A). Standard stock solutions of 1 or 10 mM were prepared in water and used for the preparation of a multicomponent standard of 0.5, 1, 5, 10, 50, 100, 500 nM and 1, 5, 10, 50 μM. Additionally, a quality control (QC) of 1 μM was prepared. Then, a dry mix including rosmarinic acid, caffeic acid, epicatechin, kaempferol, salvianolic acid B, trans-cinnamic acid, resveratrol, p-coumaric acid, tocopherol, 4-hydroxybenzoic acid, chicoric acid, and quercetine-ß-D-glucoside was dissolved in 1 mL methanol to obtain a 100 µM solution of phenolic compounds. This solution was diluted 1:10 with 5% methanol to be injected as a quality control. Furthermore, the 100 µM phenolic mixture was mixed 1:1 with a 100 µM multi-standard solution. The final stock solution had a concentration of 50 µM and was used to prepare 0.1, 0.3, 0.5, 1, 3, 5, 7.5, 10 µM solutions for external calibration.

#### 4.7.2. Liquid Chromatography High Resolution Mass Spectrometry (LC-HRMS)

The UPLC system (Vanquish Duo, Thermo Fisher Scientific, Waltham, MA, USA) was equipped with an Acquity UPLC HSS T3 column (150 × 2.01 mm, particle size 1.8 µm; Waters). Eluent A was 0.1% formic acid and eluent B was 0.1% formic acid in acetonitrile. A gradient was adapted in accordance with previous studies [69] as follows: 0–1 min isocratic 95% A, 1–19 min linear gradient starting from 95% A to 5% A, 19–22 min isocratic at 5% A and final re-equilibration from 22 to 25 min at 95% A. The flow rate was 200 µL per minute. High-resolution mass spectrometry was conducted on a Q Exactive HF quadrupole-Orbitrap mass spectrometer (Thermo Fisher Scientific) equipped with a heated electrospray (HESI) source. All samples and standards were measured by full MS (100−1500 *m*/*z*) in negative ionization mode at a resolution of 120,000. The automatic gain control (AGC) target was set to 1 × 10^6^ ions and the maximum injection time (IT) was 200 ms. The ESI source parameters were the following: sheath gas 50, auxiliary gas 14, sweep gas 3, spray voltage 2.75 kV, capillary temperature 230 °C, S-Lens RF level 45, and auxiliary gas heater 380 °C. 

#### 4.7.3. Data Processing LC/HRMS Results

Targeted data evaluation for the quantification of metabolites (organic acids, sugar, sugar phosphates, and nucleotides) was performed using Skyline software [70] with external calibration. Moreover, Thermo Scientific Compound Discoverer 2.0 software was used for metabolomics non-targeted data processing. Retention time alignment was performed within 0.15 min and 5 ppm mass tolerance. For the detection of unknown features on the MS^1^ level, a minimum peak intensity of 10,000 and 3 ppm mass tolerance was allowed. Unknown compounds were grouped according to a mass tolerance of 3 ppm and 0.15 min. Missing peaks were replaced with minimum values with a tolerance of 3 ppm and 0.1 min. A mz-Cloud search was performed with 5 ppm mass tolerance and an assignment threshold for compound annotation of 70. Limits of detection, lower limits of quantitation, and the R ^2^ value from the calibration were calculated in Skyline and in accordance to the guidelines from the Food and Drug Administration (FDA).

### 4.8. Statistical Analysis

All results are expressed as means ± SE (standard error). Experimental data were processed with XLStat software v.9.0 (Addinsoft). The effects of applied treatments were screened by one-way analysis of variance (ANOVA) with LSD’s test at 0.05 significance level for the means (*n* = 3) differences. The evaluation of LC-MS chromatograms was performed by normalizing the area of each peak by the area of the control quality and further modulating it as a function of sample fresh weight (mg). The total metabolomics dataset was subjected to Principal Component Analysis (PCA) using XLStat software v.9.0 (Addinsoft) evaluating the means of five replicates to each treatment. Significant differences in metabolome amounts as related to the studied treatments were tested by one-way ANOVA, followed by Tukey’s test (*p* < 0.05). 

## 5. Conclusions

Basil is an aromatic plant commonly used in different fields, such as foods, agriculture, nutraceutical, cosmetics, and pharmaceuticals. In the present work a joined comprehensive analytical workflow was used to investigate the effect of organic extracts from recycled agro-industrial biomasses on the synthesis and activity of plant metabolites. The assessment of plant development and biological assays combined with advanced metabolomic analyses showed a positive influence of green compost derivates on the overall plant metabolome and the production of compounds with pharmacological applicability. In particular, the application of the dissolved fractions from artichoke residues (HS-ART) and coffee grounds wastes (CT-COF) resulted in a remarkably pronounced effect on the plant biomass and yields of primary and secondary metabolites associated with a larger antioxidant and antimicrobial activity of leaves isolates. The application of HS and CTs enhanced the accumulation of aromatic amino acids such as phenylalanine, tyrosine, and tryptophan, as well as the production of saccharides involved in the production of hydroxycinnamic acids such as erythrose-4-phosphate and sedoheptulose-7-phosphate. This biological effect confirmed the synergistic effect of mixed humic extracts increasing plant metabolism. The investigation of the observed changes in the secondary metabolism of treated and untreated basil suggest an efficient activity to increase phenolic compounds such as naringenin, resveratrol, caffeic acid, and rosmarinic acid. These compounds are potent free radical scavengers/antioxidants that prevent ROS generation effectively, thus their interest is drastically growing for their important role in human health. This stimulatory effect of compost extracts could be related to the molecular composition combining structural features and the release of bioactive molecules such as lignin derivatives, phenol, and polar moieties.

The main outputs validate the valorization of agro-industrial residues as a feasible way to optimize the yield of secondary plant metabolites for nutraceutical and pharmacological applications. The preliminary characterization of organic biomasses and the advanced metabolomic investigation are unavoidable support for the targeted synthesis of beneficial natural compounds. Within a circular economy strategy, the present results further widen the potential application of sustainable technologies in nutraceutical and cosmetical fields for the increase in natural bioactive products in medicinal plants.

## Figures and Tables

**Figure 1 plants-12-00513-f001:**
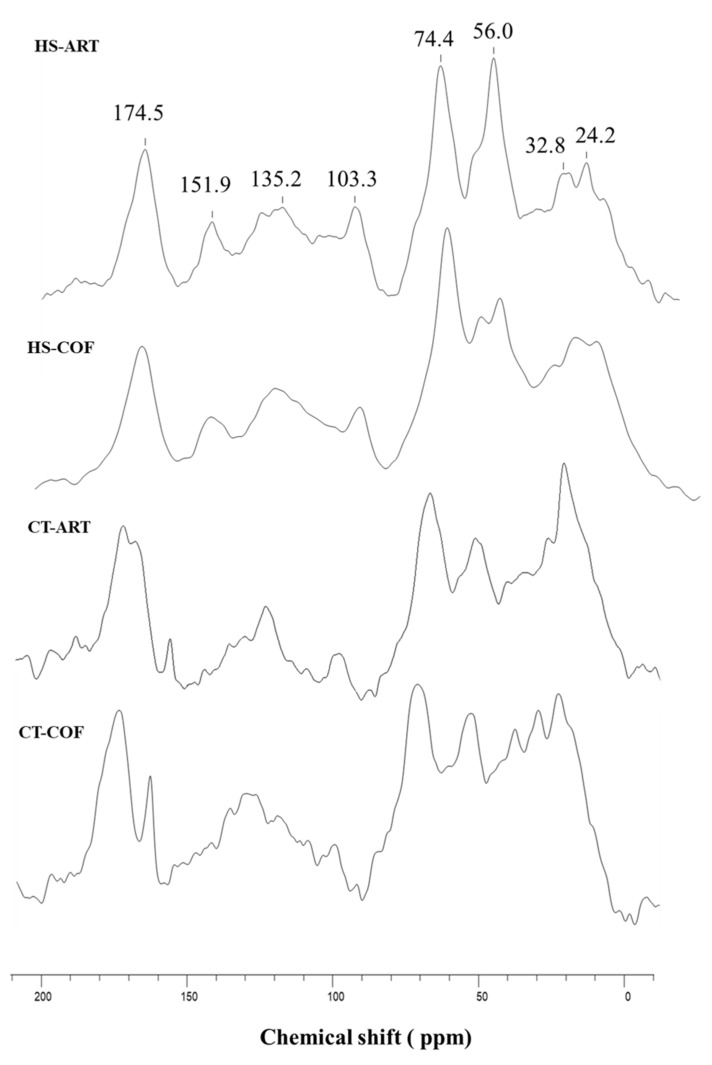
^13^C-CPMAS-NMR spectrum of humic substances and relative compost tea from artichoke and coffee composted biomasses (HS-ART, HS-COF, CT-ART, CT-COF).

**Figure 2 plants-12-00513-f002:**
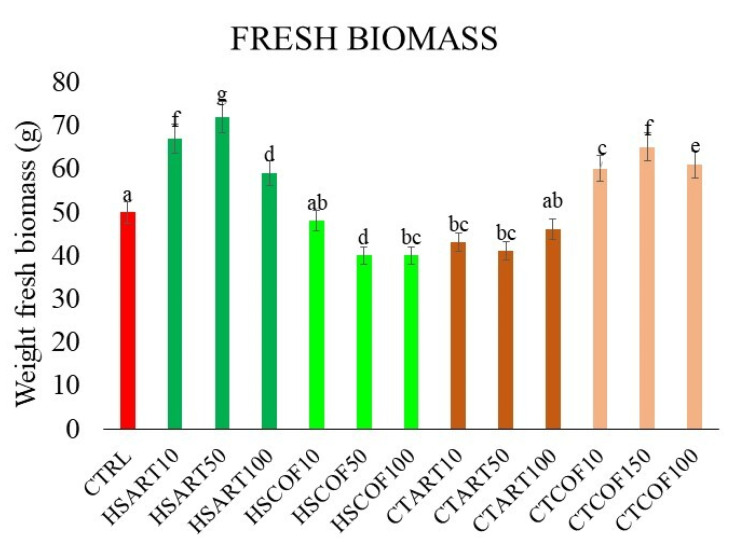
Effect of HS and CTs on weight fresh biomass of *Ocimum basilicum* plants. The different colors are associated to organic treatments applied (CTRL-red; HS-ART-green; HS-COF light green; CT-ART-brown; CT-COF orange). Vertical bars represent the standard deviation of the mean. Columns (mean ± S.D.) followed by different letters indicate significant difference according to LSD test (*p* ≤ 0.05).

**Figure 3 plants-12-00513-f003:**
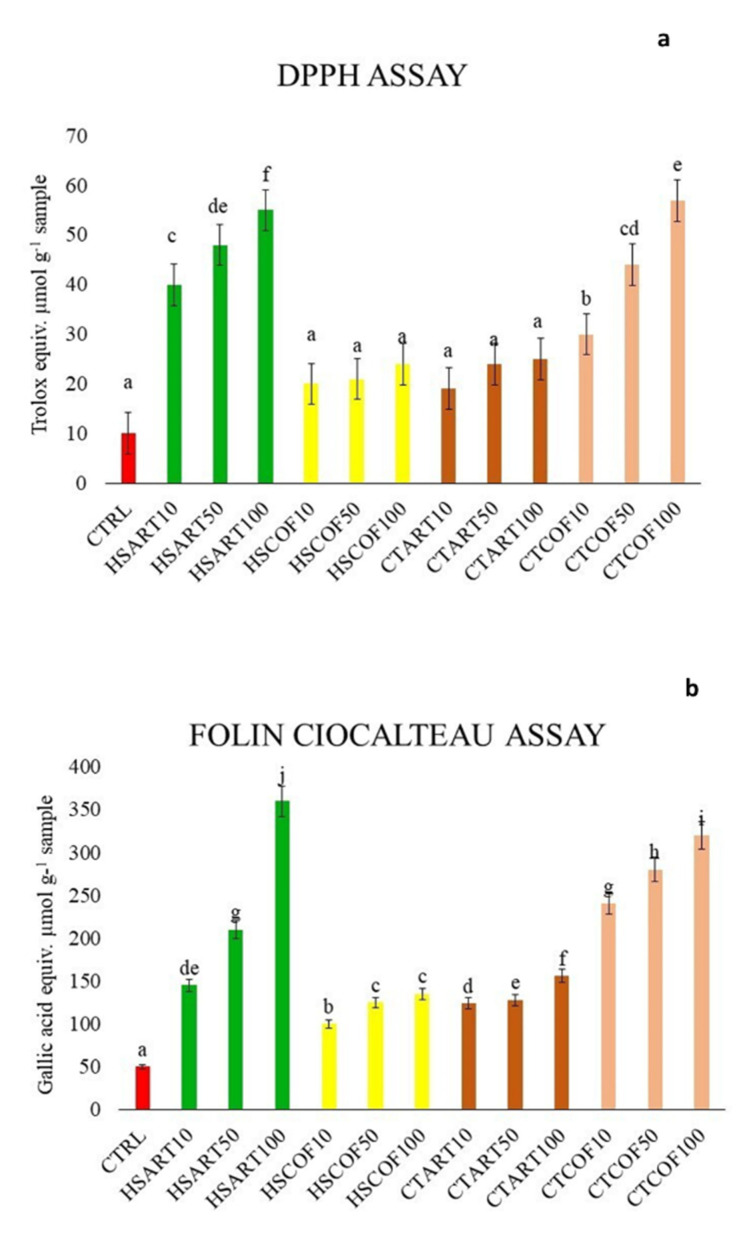
Antioxidant capacity (**a**) and Total Phenolic Content (**b**) of basil leaves from plants treated with humic substances (HS) and compost teas (CTs) extracted from artichoke and coffee composted vegetable wastes at different concentration (10–50–100 g L^−1^). The different colors are associated to organic treatments applied (CTRL-red; HS-ART-green; HS-COF yellow; CT-ART-brown; CT-COF orange). Vertical bars represent the standard deviation of the mean. Columns (mean ± S.D.) followed by different letters indicate significant difference according to LSD test (*p* ≤ 0.05).

**Figure 4 plants-12-00513-f004:**
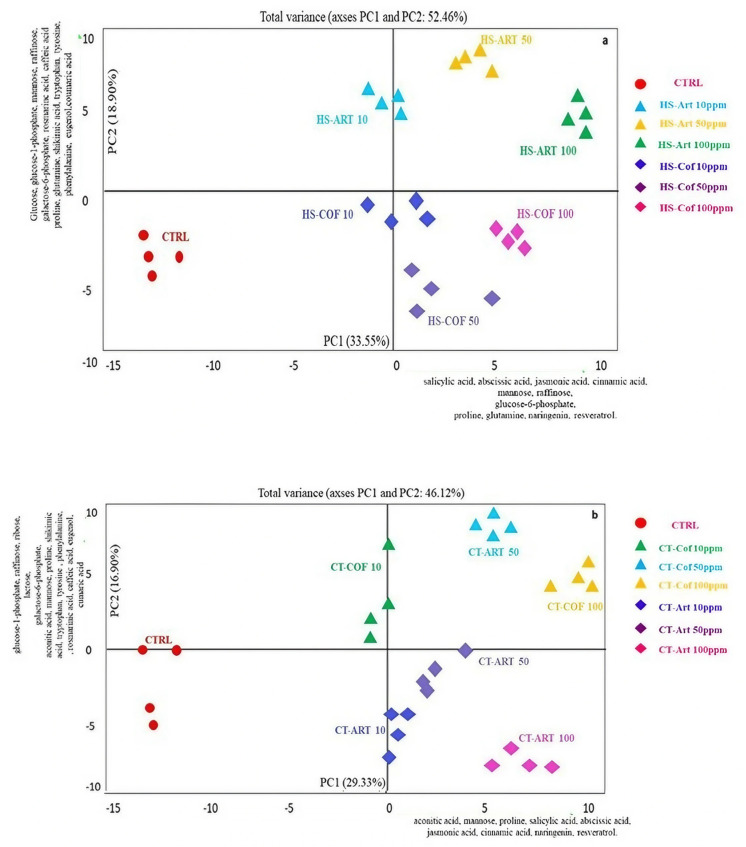
PCA score-plot based on untargeted metabolomics data obtained using Liquid Chromatography High Resolution Mass Spectrometry (LC-HRMS) of basil leaf extracts treated with humic substances (**a**) and compost teas (**b**) from artichoke and coffee grounds composted vegetable wastes (HS-ART, HS-COF, CT-ART, CT-COF) at different doses (10, 50, 100 g L^−1^). Names and directions of most significant PCA loading vectors involved in the differentiation of treatments are reported along the score-plot borders. The PCA scores represent the single value of four different biological replicates.

**Figure 5 plants-12-00513-f005:**
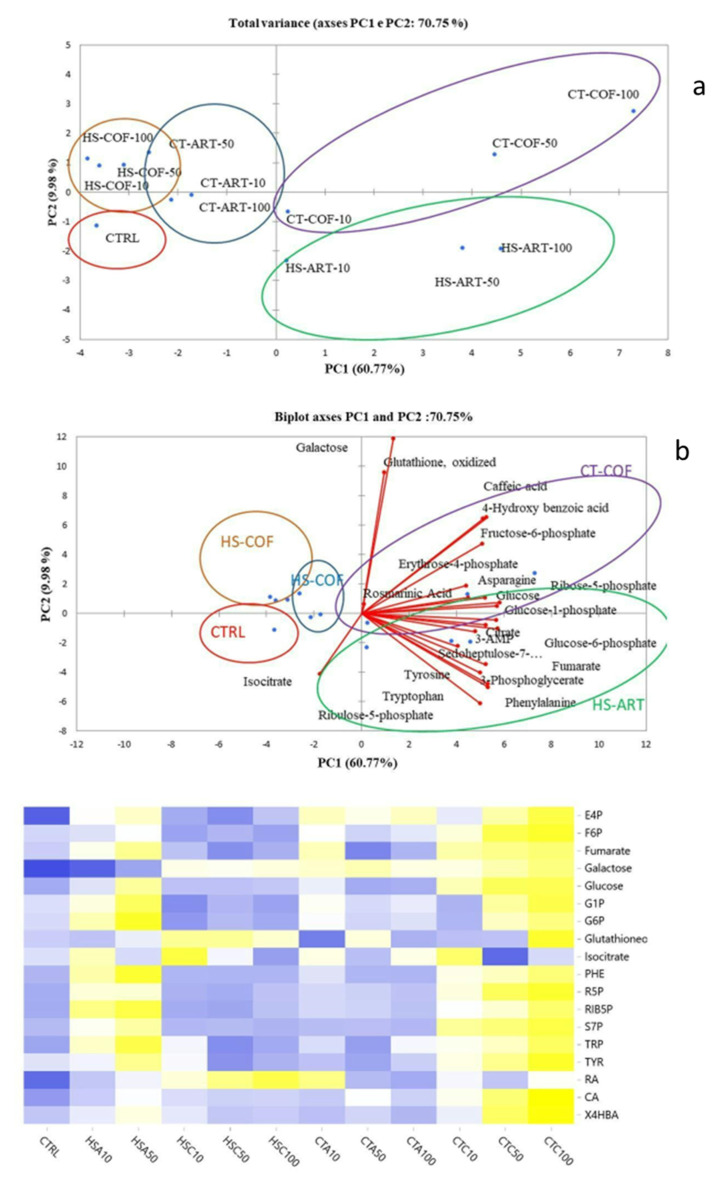
PCA biplot based on metabolites identified by targeted LC-HRMS in the apolar fraction of leaves extracts from basil plants treated with different organic extracts (**a**). Heatmap prepared from targeted metabolomics data. Each row represents a metabolite feature, and each column represents a treatment. The row Z-score or scaled expression value of each feature is plotted in yellow-green-blue-red color scale. The yellow color of the tile indicates high abundance and blue indicates low abundance. The metabolites abbreviation used are: 13PHGLY: 3-Phosphoglycerate; 3AMP: 3-AMP; ASP: Asparagine; CIT: Citrate; ERY4P: Erythrose-4-phosphate; FRU6P: Fructose-6-phosphate; FUM: Fumarate; GAL: Galactose; GLU: Glucose; GLU1P: Glucose-1-phosphate; GLU6P: Glucose-6-phosphate; GLUTOX: Glutathione, oxidized; ISOCIT: Isocitrate; PHE: Phenylalanine; R5P: Ribose-5-phosphate; RIBUL5P: Ribulose-5-phosphate; SEDEP7P: Sedoheptulose-7-phosphate; TRP: Tryptophan; TYR: Tyrosine; ROSACID: Rosmarinic Acid; CAFACID: Caffeic acid; 4HYDRBENZACID:4-Hydroxy benzoic acid. The PCA and Heatmap scores represent the average value of nine replicates (**b**).

**Figure 6 plants-12-00513-f006:**
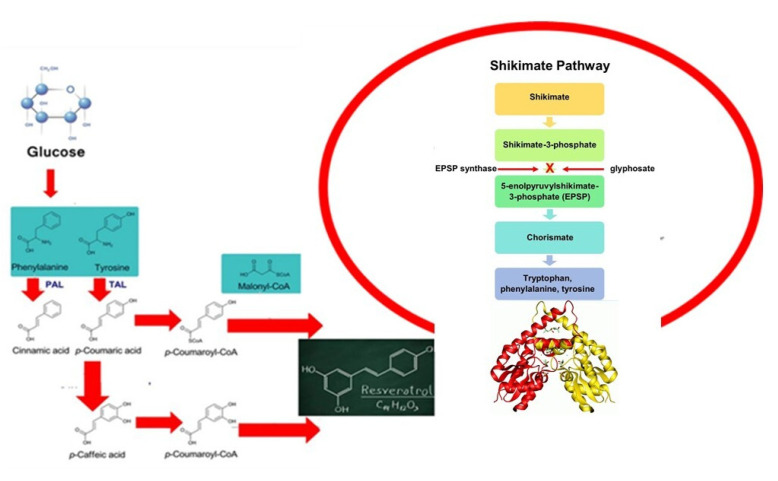
Biochemical pathways of different primary and secondary metabolites identified by nontarget and targeted LC-HRMS approaches in the apolar fraction of leaves extracts from basil plants treated with different organic extracts.

**Table 1 plants-12-00513-t001:** Relative distribution (%) of main C structures over chemical shift regions (ppm) and structural indices ^a^ of humic substances (HS) and compost teas (CT) as measured by ^13^C CPMAS-NMR spectroscopy.

Organic Extracts	Carbonyl-C (190–160)	O-Aryl-C (160–145)	Aromatic-C (145–110)	O-Alkyl-C (110–60)	CH_3_O/CN (60–45)	Alkyl-C (45–0)	HB/HI	A/OA	ARM	LigR
HS-ART	10.6	5.6	28.9	24.7	13.8	16.4	1.4	0.7	0.8	2.5
HS-COF	11.8	4	14.9	25.3	13.4	30.6	1.3	1.2	0.3	3.4
CT-ART	14.8	4.1	13.3	27.6	12.8	27.4	1.0	1.0	0.3	3.1
CT-COF	13.2	5.5	16.6	25.4	10.7	28.6	1.3	1.1	0.4	1.9

^a^ HB/HI = hydrophobicity index = [Σ(0–45) + (45–60)/2 + (110–160)/Σ(0–45) + (45–60)/2 + (60–110) + (160–190)]; A/OA = alkyl/O-alkyl ratio = [(0–45)/(60–110)]; ARM = aromaticity index [(110–160)/Σ(0–45) + (60–110)]; LigR = Lignin ratio [(45–60)/(145–160)].

**Table 2 plants-12-00513-t002:** Inhibition zones (mm) of methanolic extracts from basil plants treated with humic substances or compost teas from artichoke or coffee grounds residues (HS-ART; HS-COF; CT-ART; CT-COF) at three different concentrations (10, 50, 100 mg L^−1^) against *S. aureus*, *E. faecalis*, *E. coli*, *Pseudomonas aeruginosa*, *S. typhi* and *L. monocytogenes* by disk diffusion method (DDK).

Inhibition Zones (mm)	BSA ^2^	AMP ^3^	CTRL	HS-ART			HS-COF			CT-ART			CT-COF		
				10	50	100	10	50	100	10	50	100	10	50	100
*S. aureus*	n.i. ^1^	21.13 **	9.3	10.3 *	20.1 **	11.4 *	8.5	10.3 *	10.1	8.3	10.2 **	9.7	9.1 **	18.4 **	17.1 **
		±0.1	±0.3	±0.2	±0.3	±0.3	±0.1	±0.4	±0.2	±0.2	±0.3	±0.2	±0.2	±0.5	±0.2
*E. faecalis*	n.i.	14.32 *	8.15	8.1	11.3	12.4	7.4	7.3	7.5	7.2 *	7.2	7.1	7.6	10.6 *	9.2 *
		±0.5	±0.4	±0.4	±0.3	±0.2	±0.3	±0.1	±0.6	±0.2	±0.4	±0.7	±0.1	±0.2	±0.2
*E. coli*	n.i.	11.4	7.5	7.3	6.4	10.2	5.3	5.2	5.3	7.4	6.9	6.3	7.2 *	5.8	9.7 *
		±0.1	±0.6	±0.6	±0.6	±0.4	±0.5	±0.1	±0.3	±0.1	±0.2	±0.1	±0.4	±0.2	±0.6
*P. aeruginosa*	n.i.	13.6	8.4	8.2 **	9.6 *	9.8	8.4	8.2 *	5.7	5.2 *	5.6	6.1	7.9	9.2 *	8.7
		±0.6	±0.1	±0.1	±0.4	±0.3	±0.2	±0.6	±0.3	±0.2	±0.2 *	±0.1	±0.2	±0.7	±0.4
*S. typhi*	n.i.	18.5 *	7.3	6.4	8.3	8.6	7.4	7.9	7.3	7.9	7.4	7.1	6.70 *	8.1	7.9 *
		±0.3	±0.7	±0.5	±0.2	±0.2	±0.2	±0.2	±0.2	±0.2	±0.1	±0.7	±0.6	±0.4	±0.3
*L. monocytogenes*	n.i.	10.3	9.1	8.2	9.4	11.2 **	10.2	8.2	7.1	6.9	7.1	5.3	7.1	9.1	10.1
		±0.6	±0.4	±0.04	±0.3	±0.4	±0.3	±0.5	±0.6	±0.6	±0.6	±0.3	±0.2	±0.8	±0.02

The coefficients of variation were smaller than 6%. Asterisks indicate significant differences (* *p* < 0.05; ** *p* < 0.01) by one-way analysis of variance (ANOVA), followed by Dunnett’s method for multiple comparison. ^1^ n.i. = no inhibition, overgrowth of bacterial cells on the plates; ^2^ BSA = albumin serum bovine; ^3^ AMP ampicillin.

**Table 3 plants-12-00513-t003:** Minimal inhibition concentration (MIC) values (μg mL^−1^) of methanolic extracts from basil plants treated with humic substances or compost teas from artichoke or coffee grounds residues (HS-ART; HS-COF; CT-ART; CT-COF) at three different concentrations (10, 50, 100 mg L^−1^) against *S. aureus*, *E. faecalis*, *E. coli*, *P. aeruginosa*, *S. typhi* and *L. monocytogenes*. The coefficients of variation were invariably smaller than 6%.

MIC µg mL^−1^	BSA	AMP	CTRL	HS-ART			HS-COF			CT-ART			CT-COF		
				10	50	100	10	50	100	10	50	100	10	50	100
*S. aureus*	n.i.	1.8	2.3	1.8	1.2	1.4	2.4	2.1	1.9	2.8	2.4	3.7	2	1.9	2.1
*E. faecalis*	n.i.	2.1	2.1	1.4	1.4	1.7	2.5	2.6	2.8	2.3	2.6	3.7	1.6	1.6	1.4
*E. coli*	n.i.	2.2	2.5	1.3	1.5	1.9	2.7	2.3	3.1	3.2	2.8	3.2	1.7	1.7	1.9
*P. aeruginosa*	n.i.	3.2	2.9	1.8	2.1	2.5	2.3	2.7	2.1	3.6	2.6	2.5	1.9	1.9	2.1
*S. typhi*	n.i.	1.6	2.1	1.9	2.1	2.7	2.7	2.5	3.6	3.9	2.9	2.7	2.1	1.4	1.5
*L. monocytogenes*	n.i.	1.7	1.9	1.6	1.3	2.3	2.6	2.6	2.3	3.5	3.1	3.8	1.3	1.3	1.7

**Table 4 plants-12-00513-t004:** Concentration (nmol g^−1^) of primary and secondary metabolites from basil leaves treated with HS-ART, HS-COF, CT-ART, and CT-COF at different doses (mg L^−1^) quantified using Skyline software. Asterisks indicate significant differences (* *p* < 0.05; ** *p* < 0.01) by two-way analysis of variance (ANOVA). The metabolites reported in bold showed significant differences between plant CTRL and treatment with HS-ART and CT-COF.

		HS-ART			HS-COF			CT-ART			CT-COF		
Metabolites	Ctrl	10	50	100	10	50	100	10	50	100	10	50	100
3-Methyl-2-oxovaleric acid	0.79 ±2.1	0.93 ±4.2	1.07 ±1.5	0.76 ±2.5	0.82 ±0.1	0.73 ±2.1	0.72 ±3.9	1.20 ±1.9	0.89 ±2.1	1.04 ±0.4	1.17 ±1.1	1.07 ±1.2	0.87 ±3.1
3-Phosphoglycerate	1.16 ±1.3	1.23 ±3.6	1.43 * ±1.8	1.63 ±2.5	0.87 ** ±1.3	1.13 ±2.5	1.07 * ±2.1	1.11 ±8.1	1.19 ±2.1	1.29 ±0.1	1.24 * ±0.8	1.34 * ±1.3	1.44 ** ±1.9
3-AMP	1.26 ±0.2	1.4 * ±0.5	1.55 ** ±2.1	1.68 * ±4.2	1.22 ±2.1	1.3 6±1.9	1.47 ±2.1	1.27 ±3.1	1.36 ±0.1	1.4 ±0.6	1.33 ** ±0.2	1.46 * ±1.5	1.58 ** ±1.3
5′-Deoxy-5′-Methylthioadenosine	1.63 ±2.1	2.46 ±0.1	2.33 ±2.1	2.0 ±2.6	2.15 ±2.1	1.85 ±3.1	1.67 ±0.1	2.43 ±0.5	2.17 ±0.7	2.45 ±0.1	2.46 ±1.6	1.78 ±3.1	2.16 ±4.8
Adenine	2.62 ±2.1	2.61 ±0.6	2.49 ±0.8	2.50 ±0.9	2.22 ±0.1	2.02 ±0.8	1.82 ±0.1	2.63 ±0.9	2.37 ±0.1	2.37 ±.1.4	2.64 ±1.7	1.98 ±1.4	2.10 ±1.4
Arginine	0.98 ±3.1	2.36 ±1.2	1.16 ±1.6	0.88 ±2.1	6.44 ±2.1	1.37 ±2.7	2.28 ±3.1	3.88 ±4.1	2.28 ±1.2	6.8 ±6.1	1.6 ±0.1	1.17 ±0.4	4.43 ±0.5
Asparagine	1.8 * ±0.1	2.79 ±0.5	3.91 ±0.4	1.69 * ±0.4	2.28 ±0.6	1.99 ±0.7	1.58 * ±0.8	1.74 ±0.3	1.75 ±0.5	2.53 ±0.7	2.24 ** ±0.6	3.33 * ±2.1	4.02 * ±0.6
Aspartate	5.65 ±0.4	10.31 ±0.6	14.93 ±0.6	10.84 ±0.7	6.99 ±0.4	8.91 ±0.8	14.96 ±0.2	0.8 ±0.6	0.9 ±0.2	12.78 ±0.5	18.68 ±0.6	16.25 ±0.6	8.13 ±0.1
Cis aconitate	2.1 ±0.5	2.48 ±0.7	2.09 ±0.1	1.7 ±0.5	1.89 ±0.6	1.68 ±0.6	1.38 ±0.3	2.13 ±0.1	1.89 ±0.6	1.93 ±0.5	2.10 ±0.2	1.60 ±0.6	1.53 ±0.2
Citrate	20.68 ±0.1	24.96 ** ±0.5	26.23 * ±0.1	28.16 ±0.6 *	21.26 ±0.6	17.53 * ±0.6	22.1 ±0.9	18.94 * ±0.9	22.04 ±2.5	26.8 ±3.1	25.42 ** ±2.1	27.74 ** ±5.1	29.87 * ±0.6
Dihydroxyisovalerate	0.85 ±0.5	2.03 ±0.8	2.11 ± 0.7	2.37 ± 0.7	1.71 ± 0.3	1.81 ± 1.4	1.63 ± 1.9	1.76 ± 3.1	1.87 ± 2.1	2.30 ± 2.1	1.32 ± 0.5	1.46 ± 2.1	1.34 ± 0.2
Erythrose-4-phosphate	0.45 ± 0.2	0.66 * ± 0.7	0.72 * ± 2.1	0.86 ** ± 1.5	0.55 ± 5.1	0.51 ± 0.5	0.58 ± 2.1	0.73 ± 0.6	0.68 ± 2.1	0.73 ± 3.1	0.63 * ± 1.2	0.76 ± 1.2	0.88 * ± 0.4
Flavinadenin dinucleotide	0.33 ±2.1	0.43 ±1.4	0.34 ±2.1	0.33 ±0.3	0.33 ±0.6	0.33 ±0.1	0.30 ±1.2	0.41 ±2.1	0.42 ±0.6	0.34 ±0.7	0.43 ±2.1	0.29 ±2.1	0.36 ±0.4
Fructose-6-phosphate	1.63 ±0.1	1.72 * ±0.5	1.99 ±0.5	2.10 ±0.6	1.16 ±0.7	1.33 ±1.5	1.16 ±3.1	2.01 ±1.8	1.60 ±0.4	1.78 ±0.5	2.30 ±0.6	3.53 ±1.8	3.81 ±2.1
Fumarate	27.48 ±0.6	32.45 ** ±0.8	38.84 ±2.1	40.88 ±3.8	26.72 ±4.2	22.90 ±2.1	25.29 ±2.8	36.1 8±1.8	22.05 ±1.4	25.66 ±1.8	37.19 ±2.1	39.22 ±0.17	40.85 ±0.5
Galactose	1.06 ±1.4	1.84 ±1.7*	4.45 ±1.5	7.67 ±1.8	8.77 ±1.9	8.55 ±2.1	9.21 ±2.7	9.74 ±3.1	10.72 ±2.8	9.04 ±2.1	9.04 ±2.6	10.97 ±2.5	12.63 ±1.7
Glucose	6.97 ±0.5	9.80 ±0.3	15.93 * ±0.1	17.81 ±3.1	8.11 ±0.6*	8.11 ±2.1	8.43 ±2.1	10.44 ±1.7	6.96 ±2.1	7.16 ±5.1	14.68 * ±2.1	19.20 * ±2.6	19.49 * ±2.1
Glucose-1-phosphate	1.53 ±2.1	1.99 * ±0.4	2.89 ±1.2	1.29 * ±0.4	0.92 ±1.2	1.24 ±0.3	1.09 ±1.3	1.85 * ±1.9	1.50 ±1.2	1.63 * ±1.7	1.21 ±0.2	2.43 ±0.9	3.04 ±1.9
Glucose-6-phosphate	1.55 ±2.1	2.6 ** ±0.3	3.90 * ±0.2	1.31 ±1.3	0.90 ±2.1	1.24 * ±1.4	1.08 ±1.2	1.88 ±1.8	1.50 ±2.1	1.61 ±0.1	1.19 ±0.4	2.44 ±1.2	3.05 ±1.2
Glutathione, oxidized	2.61 ±0.2	2.46 * ±0.5	3.14 ±0.6	3.24 ±0.1	4.92 ±0.9	4.90 ±2.1	4.09 ±1.2	1.37 ±0.5	3.91 ±1.2	2.18 ±0.4	2.40 ±0.6	2.43 ±1.2	6.42 ±0.3
Glutathione, reduced	0.27 ±0.1	0.23 ±0.5	0.24 ±0.2	0.23 ±0.1	0.23 ±0.6	0.24 ±0.1	0.22 ±0.1	0.23 ±0.4	0.23 ±0.3	0.23 ±0.2	0.23 ±0.5	0.23 ±0.1	0.23 ±0.4
GMP	0.21 ±0.1	0.22 ±0.6	0.22 ±0.1	0.22 ±0.5	0.25 ±0.5	0.25 ±0.6	0.24 ±0.9	0.21 ±1.2	0.23 ±2.1	0.23 ±0.1	0.26 ±0.7	0.21 ±2.1	0.22 ±2.1
GTP	0.32 ±0.1	0.26 ±1.6	0.21 ±0.8	0.27 ±2.1	0.21 ±1.2	0.21 ±0.6	0.19 ±0.8	0.20 ±0.1	0.21 ±0.8	0.21 ±1.6	0.32 ±0.6	0.21 ±0.9	0.21 ±1.5
Histidine	3.36 ±0.1	2.72 ±2.1	4.21 ±0.1	2.43 ±0.6	3.81 ±0.7	2.96 ±1.4	2.55 ±0.5	2.45 ±0.1	2.28 ±0.7	4.30 ±2.1	2.77 ±0.7	1.29 ±2.1	3.07 ±0.6
Inosine	0.26 ±0.1	0.34 ±0.5	0.22 ±0.6	0.30 ±0.8	0.34 ±1.2	0.33 ±0.4	0.27 ±0.9	0.31 ±0.9	0.26 ±0.4	0.25 ±0.1	0.29 ±0.2	0.26 ±0.8	0.29 ±0.1
Isocitrate	18.99 ±0.1	22.71 * ±0.7	18.79 ± 0.8 **	19.96 ±0.1	27.09 * ±1.2	19.88 ± 2.1	16.46 ±0.6	21.12 ±0.1	17.71 * ±0.6	21.11 ±0.4	25.56 * ±1.2	14.33 ** ±2.1	18.75 ±0.6
Ketoisovalerate	1.07 ±1.3	1.26 ±0.5	1.39 ±1.2	1.00 ±0.6	0.95 ±1.2	0.91 ±0.6	0.88 ±1.2	1.32 ±0.6	1.07 ±0.1	1.15 ±0.5	1.33 ±0.2	1.22 ±0.7	0.99 ±0.1
Kynurenine	0.15 ± 0.1	0.15 ± 0.4	0.15 ± 0.7	0.15 ±0.4	0.16 ±1.3	0.16 ±0.7	0.14 ±0.1	0.15 ±0.4	0.15 ±0.1	0.15 ±0.4	0.14 ±0.1	0.14 ±0.1	0.15 ±0.9
Leucine	11.46 ± 1.2	16.35 ± 1.8	11.25 ±1.3	10.53 ±2.1	18.86 ±1.6	0.15 ±1.3	13.60 ±1.4	14.11 ±2.1	17.36 ±0.3	0.11 ±0.1	18.09 ±0.7	10.16 ±0.7	3.57 ±0.7
Methionine	0.86 ± 2.1	1.06 ± 1.3	0.85 ± 1.7	1.02 ±2.1	1.15 ±4.1	1.15 ±1.3	0.70 ±3.1	1.49 ±0.1	0.82 ±0.5	1.17 ±0.3	0.81 ±0.1	0.73 ±0.5	1.04 ±0.1
Mevalonic acid	4.62 ± 0.4	6.09 ± 0.1	5.52 ±0.2	4.99 ±0.4	4.30 ±0.1	4.03 ±0.1	3.65 ±0.6	5.93 ±0.1	4.46 ±0.6	4.57 ±0.1	5.19 ±0.1	4.52 ±0.2	4.03 ±0.2
N-Acetyl-L-aspartic acid	0.21 ± 0.1	0.21 ± 0.5	0.20 ±0.1	0.22 ±0.2	0.24 ±0.5	0.23 ±0.2	0.21 ±0.01	0.20 ±0.6	0.19 ±0.3	0.20 ±0.9	0.23 ±0.4	0.16 ±0.2	0.19 ±0.8
N4-Acetylcytidine	0.17 ± 0.3	0.17 ±0.02	0.17 ±0.3	0.17 ±0.16	0.17 ±0.4	0.17 ±0.5	0.16 ±0.3	0.17 ±0.2	0.17 ±0.1	0.17 ±0.2	0.17 ±0.2	0.17 ±0.4	0.17 ±0.2
NAD+	0.58 ± 0.3	0.72 ±1.5	0.65 ± 2.3	0.65 ± 0.5	0.66 ± 0.3	0.68 ± 0.3	0.56 ± 0.6	0.64 ± 0.6	0.67 ± 0.7	0.61 ± 0.3	0.63 ± 0.3	0.52 ± 0.2	0.54 ± 0.5
NADH	0.21 ± 0.2	0.21 ± 0.2	0.21 ± 0.7	0.21 ± 0.6	0.22 ± 0.7	0.21 ± 0.6	0.19 ± 0.1	0.20 ± 0.7	0.20 ± 0.2	0.21 ± 0.3	0.21 ± 0.7	0.20 ± 0.3	0.21 ± 0.2
NADP+	0.26 ± 0.3	0.26 ± 0.2	0.26 ± 1.5	0.26 ± 2.1	0.26 ± 0.4	0.26 ± 0.3	0.26 ± 0.6	0.26 ± 0.7	0.26 ± 0.3	0.26 ± 0.2	0.27 ± 0.2	0.27 ± 1.2	0.27 ± 0.6
Ornithine	1.60 ± 2.1	1.65 ± 2.4	3.73 ± 1.2	2.09 ± 0.4	2.77 ± 1.2	2.63 ± 1.6	2.61 ± 1.8	2.52 ± 1.8	2.48 ± 0.6	3.79 ± 2.1	3.30 ± 2.1	1.98 ± 1.5	1.98 ± 0.4
Phenylalanine	0.15 ± 0.1	2.39 * ± 2.1	4.07 ± 3.1	6.14 ± 1.2 **	0.14 ± 0.2	0.13 ± 1.6	0.13 ± 1.7	0.81 ± 1.2	0.14 ± 1.1	0.13 ± 0.2	1.54 ± 2.1	2.13 ** ± 1.5	3.13 * ± 0.3
Ribose-5-phosphate	0.22 ± 1.3	0.49 * ± 1.3	0.50 * ± 1.6	0.76 ± 2.1	0.23 ± 0.4	0.22 ± 0.6	0.27 ± 1.7	0.32 ± 2.1	0.30 ± 0.5	0.27 ± 0.4	0.48 ± 2.1	0.76 ± 0.5	0.86 ± 0.5
Ribulose-5-phosphate	0.23 ± 1.4	0.59 ± 2.5	0.70 ** ± 1.4	0.96 * ±1.5	0.23 ±2.1	0.22 ** ± 1.7	0.27 ± 0.6	0.31 ± 0.7	0.30 ± 1.5	0.27 * ± 0.7	0.38 * ± 0.7	0.46 * ± 0.4	0.63 ± 0.5
Sedoheptulose-7-phosphate	0.24 ± 1.5	0.75 ± 1.2	1.20 ± 2.6	1.72 ± 3.2	0.23 ** ± 3.6	0.25 ± 1.4	0.20 ± 0.2	0.26 ± 0.6	0.27 ± 0.8	0.23 ± 0.3	1.26 ± 0.3	1.45 ± 0.3	1.76 ± 0.1
Seleno-methionine	0.12 ± 2.5	0.11 ± 1.6	0.11 ± 1.7	0.12 ± 1.8	0.12 ± 2.1	0.12 ± 2.1	0.12 ± 1.4	0.12 ± 1.2	0.12 ± 0.6	0.12 ± 1.5	0.12 ± 1.8	0.12 ± 2.6	0.12 ± 2.6
Serine	4.37 ± 0.4	3.59 ± 0.6	3.98 ± 0.7	4.45 ± 0.7	3.80 ± 0.7	5.70 ± 0.8	3.48 ± 0.8	3.75 ± 0.9	3.82 ± 0.9	4.6 1 ± 0.5	4.83 ± 0.6	2.3 5 ± 0.6	3.35 ± 0.8
Threonine	0.82 ± 0.8	0.87 ± 0.7	0.90 ± 0.3	0.83 ± 1.5	0.93 ± 0.7	0.77 ± 0.3	0.68 ± 0.6	0.95 ± 0.7	0.78 ± 0.4	0.94 ± 0.3	1.04 ± 2.1	0.73 ± 1.7	0.82 ± 1.8
Thymidine	0.16 ± 1.4	0.18 ± 1.6	0.17 ± 1.8	0.17 ± 1.5	0.18 ± 2.1	0.17 ± 1.8	0.15 ± 1.2	0.17 ± 1.8	0.17 ± 0.5	0.17 ± 0.6	0.18 ± 0.7	0.16 ± 0.4	0.17 ± 0.2
Thymine	0.22 ± 1.7	0.25 ± 1.5 *	0.23 ± 1.6	0.23 ± 2.1	0.24 ± 2.7	0.23 ± 1.8	0.22 ± 1.6	0.24 * ± 1.5	0.24 ± 1.2	0.24 ± 1.4	0.26 ± 1.7	0.26 ± 1.6	0.26 ± 1.2
Tryptophan	18.65 ± 2.1	33.77 ** ± 1.5	46.50 ** ± 1.7	53.99 ± 1.6 *	27.44 ± 0.4 *	15.97 ± 0.5	19.16 ± 1.4	24.68 ± 1.4	17.90 ± 0.5	27.47 ± 0.7	30.04 * ± 0.2	34.35 * ± 0.6	44.25 ± 0.2
Tyrosine	8.66 ± 0.3	9.49 ± 0.6	15.27 ± 0.3	17.32 ± 0.3	9.85 ± 0.2	4.69 ± 0.3	6.29 ± 0.6	7.03 ± 0.5	5.61 ± 2.1	7.68 ± 1.3	11.32 ± 1.7	15.22 ± 1.8	19.96 ± 2.7
Valine	0.06 ± 0.2	5.82 ± 0.7	4.40 ± 0.8	4.71 ± 0.5	5.10 ± 0.7	5.10 ± 0.7	7.32 ± 0.4	8.47 ± 0.6	4.94 ± 0.7	0.14 ± 0.3	8.16 ± 0.8	8.19 ± 0.5	4.83 ± 0.7
Xanthine	0.14 ± 2.1	0.15 ± 4.1	0.17 ± 0.7	0.19 ± 1.6	0.23 ± 2.5	0.18 ± 2.8	0.14 ± 1.6	0.16 ± 1.8	0.12 ± 1.6	0.16 ± 2.1	0.16 ± 4.1	0.16 ± 2.6	0.13 ± 2.5
Rosmarinic Acid	9.57 * ± 1.6	12.85 * ± 1.8	14.45 ± 1.7	19.31 ** ± 1.7	16.11 ± 2.5	18.88 * ± 2.9	20.96 ± 2.5	18.84 ± 2.4	12.39 ± 2.1	11.85 ± 4.6	14.59 ± 2.4	12.81 * ± 1.5	15.15 ** ± 1.5
Caffeic acid	21.33 ± 2.1	27.33 ± 4.1	32.94 ± 1.7*	43.96 ± 1.8°	29.11 ± 2.7	27.75 ± 1.7	28.20 ± 2.1	27.00 ± 2.6	33.06 ** ± 1.7	27.74 ± 1.4	34.77 ± 1.6	47.83 ± 0.5	60.37 * ± 0.5
4-Hydroxy benzoic acid	2.18 ± 4.1	2.96 ± 0.1	3.94 ± 0.2	3.5 2 ± 1.5	2.9 8 ± 1.3	2.20 ± 1.7	2.30 ± 1.1	2.03 ± 0.9	2.49 ± 2.1	1.85 ± 1.5	3.04 ± 1.4	6.43 ± 1.8	8.24 ± 1.2

## Data Availability

The data summary is contained within the article. NMR and MS raw data are available upon request.

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
