# Peer review of "Evaluation of Sustainable Recycled Products to Increase the Production of Nutraceutical and Antibacterial Molecules in Basil Plants by a Combined Metabolomic Approach"

_plants, 2023, doi:10.3390/plants12030513_

Round 1
Reviewer 1 Report
Dear authors,
The manuscript entitled "Evaluation of sustainable recycled products to increase the production of nutraceutical molecules in basil plants by combined metabolomic approach" to optimized the yield of specific metabolites with nutraceutical features by applying eco-friendly methodologies. It presents scientific relevance for the area of Chemistry, Natural products and Food area. Most authors have published articles related to the theme of the manuscript found in databases (Sciencedirect, Pubmed, MDPI, Web of science, etc). The language (English) is satisfactory (but, I suggest the final revision)! However, you need to change some details/information in the Title, Abstract, Introduction, Material and Methods, results and discussion, and conclusions.
Abstract: Adequate! But:
- What do the authors call "eco-friendly methods"? The authors used chloroform and acetonitrile in the methods!!! Solvents that go against the principles of green chemistry! I suggest review of this term!
- In results, the authors wrote “…, as well as their antimicrobial activity”. There is no information in the title or in other parts of the abstract about antimicrobial activity. I suggest review!
- The abstract is well written, with details of the methods used. However, I suggest reducing the information on "methods" and inserting the results obtained (numerical data) more relevant.
- The keywords “Recycling biomasses”, “pharmacological activity” and “antioxidant activity” do not appear in the title and/or abstract. I suggest review! Is the term “pharmacological activity” appropriate for this manuscript? Have in vivo assays been performed? "Antioxidant activity” or "Antioxidant capacity"?
1. Introduction section:
- As an article proposal is the development and validation of the analytical method, I suggest including information on analytical techniques and more references on the determination of the species studied in similar matrices by liquid chromatography (LC). I suggest adding the following references:
* Evaluation of multielement/proximate composition and bioactive phenolics contents of unconventional edible plants from Brazil using multivariate analysis techniques. Food Chemistry (2021). https://doi.org/10.1016/j.foodchem.2021.129995
* Multielementar/centesimal composition and determination of bioactive phenolics in dried fruits and capsules containing Goji berries (Lycium barbarum L.). Food Chemistry (2019). https://doi.org/10.1016/j.foodchem.2018.05.124
- I suggest at the end of the introduction, to highlight the "innovative" proposal of the method, as well as the advantages / disadvantages.
2. Results
Wouldn't it be more interesting to combine the "results” with the "discussion" to better describe the findings and compare them with other works published in the literature? I suggest expanding the discussions!
- Page 3, in “2.1. Molecular characterization of compost extracts” section: The text is written in a single paragraph! I suggest 2 or 3 paragraphs!
- Page 5, in “2.2. Phenological parameter of Basil plants” section: What do the colors in Figure 2 represent?
- Page 6, in “2.3. Antioxidant activity of basil extracts” section: "Antioxidant activity” or "Antioxidant capacity"? I suggest replacing by "Antioxidant capacity" and reviewing throughout the manuscript. What do the colors in Figure 3 represent?
- Pages 7-8: I suggest indicating a legend (or in the title) of Tables 2 and 3, with the meaning of the acronyms!
- Page 12: The text is written in a single paragraph! I suggest 2 or 3 paragraphs!
- Page 12: Analyte concentrations (Table 4) are expressed (nmol g-1). Wouldn't it be more interesting in a unit (mass/mass)?
- Pages 13-14: To replace “Table 5. Continue” by “Table 4. Continue”.
- Pages 12-14: In Table 4, for some metabolites (for example, 3-Methyl-2-oxovaleric acid; 3-Phosphoglycerate...and others) some data related to the "standard deviation" are higher than the determined concentration. How can this be? These concentrations are probably below the limits of detection and quantification, associated with the method/analytical technique used! I suggest reviewing the calculations of analyte concentrations!
3. Discussion
Wouldn't it be more interesting to combine the "results” with the "discussion" to better describe the findings and compare them with other works published in the literature? I suggest expanding the discussions!
- Page 15, in “3.1.1. Plant biomass” section: Long paragraph! I suggest splitting between them!
- Page 16, in “3.1.3. Antimicrobial activity”: I suggest expanding the discussions!
- Page 18: The text is written in a single paragraph! I suggest 2 or 3 paragraphs!
- I suggest, at the end of the "results and discussion", to write a paragraph summarizing the findings and their impacts on the research proposal.
4. Material and Methods
- The methodological proposals are coherent, but it needs to improve, especially in the analytical aspects.
- Pages 18-19, in “4.1. Humic substances (HS) and compost teas (CTs)” section: How long were the samples stored until the time of analysis? Only seven days?
- Page 19: to replace “ml” by “mL” and throughout the manuscript.
- Page 19, in “4.2. Experimental design, plant growth, sampling, and analyses” and “4.3. Extraction of plant leaf metabolites” section: Did the procedure follow any previous protocol/manuscript? If yes, indicate the reference!
- Page 20: In addition to the determination of Total phenolic compounds (TPC), was the total content of flavonoids performed?
- Page 22, in “4.7.2. Liquid Chromatography High Resolution Mass Spectrometry (LC-HRMS) and “4.7.3. Data processing LC/HRMS results” section: Which concentration ranges were studied? What are the analytical validation parameters used? Has the proposed method been validated? If so, which protocol / guidelines did you follow? What are the validation parameters studied? Precision, accuracy, LOD, LOQ, robustness, etc. What concentration levels are used to assess accuracy? I suggest detailing the proposed method in more detail...
5. Conclusion: Adequate, but I suggest to indicate disadvantages/limitations of the method and the study!
Table and Figures: Adequate. I suggest checking the suggestions throughout the review.
Supplementary material: Adequate.
References: Please, check if the references are in accordance with the journal's rules.
Author Response
Reviewer #1
R1: The manuscript entitled "Evaluation of sustainable recycled products to increase the production of nutraceutical molecules in basil plants by combined metabolomic approach" to optimized the yield of specific metabolites with nutraceutical features by applying eco-friendly methodologies. It presents scientific relevance for the area of Chemistry, Natural products and Food area. Most authors have published articles related to the theme of the manuscript found in databases (Sciencedirect, Pubmed, MDPI, Web of science, etc). The language (English) is satisfactory (but, I suggest the final revision)! However, you need to change some details/information in the Title, Abstract, Introduction, Material and Methods, results and discussion, and conclusions.
Abstract: Adequate! But:
What do the authors call "eco-friendly methods"? The authors used chloroform and acetonitrile in the methods!!! Solvents that go against the principles of green chemistry! I suggest review of this term!
A: We modified the statement in accordance with the Reviewer’s indication as follow: Lines 18-21 “In this study, humic substances (HS) and compost teas (CTs) extracted from artichoke (ART), and coffee grounds (COF) recycled biomasses were employed on Ocimum Basilicum plants to optimize the yield of specific metabolites with nutraceutical and antibacterial features by applying sustainable strategies.”
R1: In results, the authors wrote “…, as well as their antimicrobial activity”. There is no information in the title or in other parts of the abstract about antimicrobial activity. I suggest review!
A: We modified the title as it follows: “Evaluation of sustainable recycled products to increase the production of nutraceutical and antibacterial molecules in basil plants by combined metabolomic approach”.
The Abstract has been modified accordingly: line 21 (please refer to the previous point)
R1: The abstract is well written, with details of the methods used. However, I suggest reducing the information on "methods" and inserting the results obtained (numerical data) more relevant.
A: We modified text as it follows: “HS-ART and CT-COF influenced antioxidant activity of basil metabolites (TEAC values beetween 39 and 55 mmol g -1) and their antimicrobial efficacy (MIC value between 3.7 and 1.3 µg mL-1” ( Lines 25-27).
R1: The keywords “Recycling biomasses”, “pharmacological activity” and “antioxidant activity” do not appear in the title and/or abstract. I suggest review! Is the term “pharmacological activity” appropriate for this manuscript? Have in vivo assays been performed? "Antioxidant activity” or "Antioxidant capacity"?
A: We modified keywords as it follows: “Recycling biomasses; metabolomics; phenolic compounds; antioxidant capacity; antibacterial properties”( Lines 34-36)”
R1: Introduction section:
- As an article proposal is the development and validation of the analytical method, I suggest including information on analytical techniques and more references on the determination of the species studied in similar matrices by liquid chromatography (LC). I suggest adding the following references:
* Evaluation of multielement/proximate composition and bioactive phenolics contents of unconventional edible plants from Brazil using multivariate analysis techniques. Food Chemistry (2021). https://doi.org/10.1016/j.foodchem.2021.129995
* Multielementar/centesimal composition and determination of bioactive phenolics in dried fruits and capsules containing Goji berries (Lycium barbarum L.). Food Chemistry (2019). https://doi.org/10.1016/j.foodchem.2018.05.124
A: Additional information on analytical techniques have been included (please see the subsequent points). We added the suggested references (lines 96, 860-867). The references have been renumbered accordingly in the text and in the final list.
R1: I suggest at the end of the introduction, to highlight the "innovative" proposal of the method, as well as the advantages / disadvantages.
A: We modified the text as it follows (lines 103-108): “The advantage of this strategies is related to the use of recycling biomasses to increase the production of antioxidant and antimicrobial natural compounds. Conversely, possible drawbacks are related to the reproducibility of composting processes due to the seasonal variation of available agro-industrial wastes. To face this issue the on-farm composting methodology may represents an easy-to-handle solution for the careful check of fresh biomasses and composting conditions”
R1: Wouldn't it be more interesting to combine the "results” with the "discussion" to better describe the findings and compare them with other works published in the literature? I suggest expanding the discussions!
A: We would like to thank you for this comment, aimed at improving the presentation of the study in a more critical way. The split or integration of these two sections is always matter of debate, depending mainly on the approach and way of thinking of different researchers. In our opinion, since our study include different integrated analytical methodologies, we believe that the split between Results and Discussion is helpful to elucidate the extensive amount of experimental data thereby facilitating the comprehensive understanding and explanation of background, objectives, and outputs. The combination of data and comments in one section may produce an unfruitful repetition of concepts, explanatory statements and cross-references between various paragraphs thereby making the manuscript dull reading and muddled.
R1: Page 3, in “2.1. Molecular characterization of compost extracts” section: The text is written in a single paragraph! I suggest 2 or 3 paragraphs!
A: Since the molecular characterization is based on a unique technique (solid state NMR) the split in 2 or 3 sections would be ineffective in the description and correlation of analysed parameters. Although we analyzed two different compost extracts (i.e. HS and CTs) the synthetic and simultaneous comparison of structural data highlight the importance of specific conformational and molecular characteristics for the comprehensive understanding of structural-activity relationship.
R1: Page 5, in “2.2. Phenological parameter of Basil plants” section: What do the colors in Figure 2 represent?
A: The different colors exhibit in Figure 2 are related to the different organic treatments. We modified the description of Figure 2 as it follows: “Effect of HS and CTs on weight fresh biomass of Ocinum basilicum plants. The different colors are associated to organic treatments applied (CTRL- red; HS-ART – green; HS-COF light green; CT-ART- brown; CT- COF orange). Vertical bars represent the standard deviation of the mean. Columns (mean ± S.D.) followed by different letters indicate significant difference according to LSD test (p ≤ 0.05).”
R1: Page 6, in “2.3. Antioxidant activity of basil extracts” section: "Antioxidant activity” or "Antioxidant capacity"? I suggest replacing by "Antioxidant capacity" and reviewing throughout the manuscript. What do the colors in Figure 3 represent?
A: We replaced the antioxidant activity as antioxidant capacity in all text. The colors in Figure 3 are related to the different organic treatments. We modified the description of Figure 3 as it follows: “Antioxidant capacity (panel a) and Total Phenolic Content (panel b) of basil leaves from plants treated with humic substances (HS) and compost teas (CTs) extracted from artichoke and coffee composted vegetable wastes at different concentration (10-50-100 g L-1). The different colors are associated to organic treatments applied (CTRL- red; HS-ART – green; HS-COF yellow; CT-ART- brown; CT- COF orange). Vertical bars represent the standard deviation of the mean. Columns (mean ± S.D.) followed by different letters indicate significant difference according to LSD test (p ≤ 0.05).”
R1: Pages 7-8: I suggest indicating a legend (or in the title) of Tables 2 and 3, with the meaning of the acronyms!
A: Thank you for your indication. We explained the acronyms of different treatments in the title of Tables 2 and 3.
R1: Page 12: Analyte concentrations (Table 4) are expressed (nmol g-1). Wouldn't it be more interesting in a unit (mass/mass)?
A: We expresses the metabolites concentrations as nmol g-1 as required by Skyline software employed to target evaluation analysis.
R1: Pages 13-14: To replace “Table 5. Continue” by “Table 4. Continue”.
A: We modified the text.
R1: Pages 12-14: In Table 4, for some metabolites (for example, 3-Methyl-2-oxovaleric acid; 3-Phosphoglycerate...and others) some data related to the "standard deviation" are higher than the determined concentration. How can this be? These concentrations are probably below the limits of detection and quantification, associated with the method/analytical technique used! I suggest reviewing the calculations of analyte concentrations!
- We would like to thank you for this comment, but our analyte concentration is included in the limits of detection and quantification applied as reported as it follows: “LODs ranged between 0.03 and 0.9 µM for 90% of the
compounds (LODs of 3-phosphoglycerate, glutathione, NADP+,
S-Adenosyl-homocysteine, citrate, iso-citrate, dAMP and dCMP were higher
than 0.9 µM). LLOQs ranged between 1 and 5 µM accordingly (Lines 328-332 )”.The larger standards deviation is generally related to biological samples that exhibit a higher variability.
R1: 3. Discussion Wouldn't it be more interesting to combine the "results” with the "discussion" to better describe the findings and compare them with other works published in the literature? I suggest expanding the discussions!
A: Please see our reply for the previous comment about Results.
About the extension of Discussion section, we have slightly extended it (lines 447-453) with a split in different subsections to improve the readability and the comprehensive correlation between analytical approaches and acquires outputs. Moreover, it has to be considered that part of the comments have been unavoidably included in the Results section, and that currently the Discussion already represents about the 40% of the manuscript together with Introduction and Results sections
R1: Page 15, in “3.1.1. Plant biomass” section: Long paragraph! I suggest splitting between them!
- The comments on the outputs of the plant analyses have been already split in three subsections; a further split in our opinion will result redundant, also for the split of the subsequent paragraphs in the Discussion (please see next points)
R1: Page 16, in “3.1.3. Antimicrobial activity”: I suggest expanding the discussions!
A: We modified the text as it follows (lines 447-454): “Generally, leaves of basil, either fresh or dried, have a large amount of antioxidant aromatic compounds. Therefore, basil extract could be used as a natural preservative to extend the shelf life of perishable foods according to the UK Food Standards Agency [57]. The antioxidant features and immune-boosting properties of basil have been also associated to observed protection against Helicobacter pylori, a bacterial strain in-volved in chronic gastric ulcers [58]. Different processes can affect the antimicrobial activity of aromatic plant extracts such as geographical origin, plant quality, method of extraction, and the application of organic fertilizers [59].”
R1: Page 18: The text is written in a single paragraph! I suggest 2 or 3 paragraphs!
A: Thank you for this suggestion; it will improve the coherent readability and understanding of the Discussion section. We have split the paragraph 3.2. Combined Untarget and Target metabolomic approaches in two different paragraphs 3.2.1” Effect of organic derivates on plants biochemical pathways to produce metabolites by nutraceutical application” and 3.2.2.” Effect of compost derivates on plants phenylpropanoid derivatives”.
R1: I suggest, at the end of the "results and discussion", to write a paragraph summarizing the findings and their impacts on the research proposal.
A: Differently from the previous point, we do not agree with this indication. The manuscript provides extended sections for Results and Discussion; an additional summary, before the conclusion, would be repetitive and overloading the text without improving the scientific appraisal
R1: The methodological proposals are coherent, but it needs to improve, especially in the analytical aspects. Pages 18-19, in “4.1. Humic substances (HS) and compost teas (CTs)” section: How long were the samples stored until the time of analysis? Only seven days?
A: We reported in materials and methods section that the different humic substances and compost teas samples have been freeze dried and stored a 4 °C to further analysis.
R1: Page 19: to replace “ml” by “mL” and throughout the manuscript.
A: Thank you for your advice. We replaced ml by mL in the manuscript.
R1: Page 19, in “4.2. Experimental design, plant growth, sampling, and analyses” and “4.3. Extraction of plant leaf metabolites” section: Did the procedure follow any previous protocol/manuscript? If yes, indicate the reference!
A: We added the relative reference.
R1: Page 20: In addition to the determination of Total phenolic compounds (TPC), was the total content of flavonoids performed?
A: We did not perform flavonoid content evaluation but focused our attention on total phenol content to correlate these results with antioxidant and antimicrobial activity of basil plants.
R1: Page 22, in “4.7.2. Liquid Chromatography High Resolution Mass Spectrometry (LC-HRMS) and “4.7.3. Data processing LC/HRMS results” section: Which concentration ranges were studied? What are the analytical validation parameters used? Has the proposed method been validated? If so, which protocol / guidelines did you follow? What are the validation parameters studied? Precision, accuracy, LOD, LOQ, robustness, etc. What concentration levels are used to assess accuracy? I suggest detailing the proposed method in more detail...
A: Thank you for the comments. Concentrations ranges were 0.1 nM to 10 µM and retention time, m/z in MS1 smaller than 5 ppm for the panel of commercially available standards (Lines 734-739). The LC-MS method was adapted from our validated and benchmarked non-targeted metabolomics RP-MS method (reference number 70) using an established gradient system from the literature (reference number 69). Additionally, limits of detection (LODs), lower limits of quantitation (LLOQs) and R2 from the calibration were calculated in Skyline and in accordance to the guidelines from the Food and Drug Administration (FDA) (Lines 737-739). In detail, 101 metabolites were quantified using external standards in a range from
0.1 to 10 µM. LODs ranged between 0.03 and 0.9 µM for 90% of the
compounds (LODs of 3-phosphoglycerate, glutathione, NADP+,
S-Adenosyl-homocysteine, citrate, iso-citrate, dAMP and dCMP were higher
than 0.9 µM). LOQs ranged between 1 and 5 µM accordingly (Lines 328-332)
R1: 5. Conclusion: Adequate, but I suggest to indicate disadvantages/limitations of the method and the study!
A: The Conclusion has been slightly modified according with the Reviewer’s suggestion (lines 783-792)
Reviewer 2 Report
1. For more convenience of reviewing, please add the continuous number for each line of the whole manuscript for reviewers and editors.
2. Please add one “period” at the end of the last paragraph of Abstract.
3. Section 2.1, paragraph 2, line 3, please remove the “,” before “(Table 1)”.
4. Table 1 title, Line 2, the “C” in “Compost” doesn’t need to be capital. Please revise it.
5. Please add “a” “b” panels labeling in Figure 3. The same with Figure 4 and Figure 5.
6. In Figure 5b, do the different colors mean different concentration of metabolites? Please specify the correlation of values and the colors.
7. The tables below Table 4 are “Table 4 continue” or “Table 5 continue”? Table 5 is not cited in the main text. Please double check the description.
8. Figure 6, the shikimic pathway is not completely showed in the figure, the left side is hided and showed half in the figure.
9. The references format is not consistent, please double check it and cite the references precisely.
Author Response
R2: For more convenience of reviewing, please add the continuous number for each line of the whole manuscript for reviewers and editors.
A: Thank you for the comment, we added continuous number for each line.
R2: Please add one “period” at the end of the last paragraph of Abstract.
A: We have substantially changed the Abstract
R2: Section 2.1, paragraph 2, line 3, please remove the “,” before “(Table 1)”.
A: We modified the text removing “” before Table 1.
R2: Table 1 title, Line 2, the “C” in “Compost” doesn’t need to be capital. Please revise it.
A: We modified the capital letter in the table headline accordingly.
R2: Please add “a” “b” panel labeling in Figure 3. The same with Figure 4 and Figure 5.
A: We modified the Figure 3.
R2: In Figure 5b, do the different colors mean different concentration of metabolites? Please specify the correlation of values and the colors.
A: We modified the text as it follows: “The row Z-score or scaled expression value of each feature is plotted in yellow-green-blue-red color scale. The yellow color of the tile indicates high abundance and blue indicates low abundance”.
R2: The tables below Table 4 are “Table 4 continue” or “Table 5 continue”? Table 5 is not cited in the main text. Please double check the description.
A: We remove the mistake and corrected the Table 4 continue in the text.
R2: Figure 6, the shikimic pathway is not completely showed in the figure, the left side is hided and showed half in the figure.
A: We modified the Figure 6.
R2: The references format is not consistent, please double check it and cite the references precisely.
A: We have adjusted the references format.
Reviewer 3 Report
Interesting paper . I find that should be publishef . The approach is otiginal . The recycle of organic products to increase the nutraceutical composition in cultivatef plants.
This is in the field of both the herbal medicinìne and susteinability
I believe that the only unappropriate references was “ as inhibitor of carcinogenesis” once cited and never treated or discussed.
The use of CP MAS NMR is well applied and gave good results.
Two spelling errors Nuclear Mag-netic Resonance instead of Nuclear Mag-netic.
The only perplexity about the term used “Biostimulant”.
Cite “In this study, natural organic compounds from recycled agricultural biomasses have been employed as biostimulants on basil plants to improve the synthesis of specific mol-ecules with nutraceutical or medical application”.
And the sentence
“as tailored biostimulant for the biosynthesis”
And the conclusion “The cutting-edge challenge is the potential definition of viable protocol to correlate the characteristics of recycled natu-ral organic biostimulants to yield and efficacy of specific plant products with pharmaceu-tical application.”
I believe that instead of biostimulation the correct definition is the term “effector” able to address the metabolism to different metabolic pathways and activating different substrates
Correctly written here
“The investigation on the ob-served changes in the secondary metabolism of treated and untreated basil suggest an efficient activity to increase phenolic compounds such as naringenin, resveratrol, caffeic acid or rosmarinic acid. These compounds are potent free radical scavengers/antioxidants that prevent ROS generation effectively, thus their interest is drastically growing for their important role in human health. This metabolic addressing effect of compost extracts could be re-lated to the molecular composition combining structural eatures and the release of bioac-tive molecules such as lignin derivatives, phenol and polar moieties.
Author Response
Reviewer #3
Interesting paper. I find that should be published. The approach is original. The recycle of organic products to increase the nutraceutical composition in cultivated plants.
This is in the field of both the herbal medicine and sustainability
Thank you for your positive comments and your kind appraisal
R3 I believe that the only inappropriate references was “ as inhibitor of carcinogenesis” once cited and never treated or discussed.
A The statement was modified in the revised manuscript (lines 50-51)
R3 The use of CP MAS NMR is well applied and gave good results.
A Thank you for your appreciation
R3 Two spelling errors Nuclear Magnetic Resonance instead of Nuclear Magnetic.
A Thank you for your check; the terms have been corrected (line 110)
R3 The only perplexity about the term used “Biostimulant”.
Cite “In this study, natural organic compounds from recycled agricultural biomasses have been employed as biostimulants on basil plants to improve the synthesis of specific molecules with nutraceutical or medical application”.
And the sentence: “as tailored biostimulant for the biosynthesis”
And the conclusion “The cutting-edge challenge is the potential definition of viable protocol to correlate the characteristics of recycled natural organic biostimulants to yield and efficacy of specific plant products with pharmaceutical application.”
I believe that instead of biostimulation the correct definition is the term “effector” able to address the metabolism to different metabolic pathways and activating different substrates
Correctly written here
“The investigation on the observed changes in the secondary metabolism of treated and untreated basil suggest an efficient activity to increase phenolic compounds such as naringenin, resveratrol, caffeic acid or rosmarinic acid. These compounds are potent free radical scavengers/antioxidants that prevent ROS generation effectively, thus their interest is drastically growing for their important role in human health. This metabolic addressing effect of compost extracts could be related to the molecular composition combining structural features and the release of bioactive molecules such as lignin derivatives, phenol, and polar moieties.
A The definition of Prof Du Jardin state the biostimulant as follows: “A plant biostimulant is any substance or microorganism applied to plants with the aim to enhance nutrition efficiency, abiotic stress tolerance and/or crop quality traits, regardless of its nutrient content”. (du Jardin, P. (2015) Plants Biostimulants: Definition, Concept, Main Categories and Regulations. Scientia Horticulturae, 196, 3-14 https://doi.org/10.1016/j.scienta.2015.09.021)
Furthermore, the European biostimulants industry council (EBIC, 2012) defined plant biostimulants as “containing substance(s) and/or micro-organisms whose function when applied to plants or the rhizosphere is to stimulate natural processes to enhance/benefit nutrient uptake, nutrient efficiency, tolerance to abiotic stress, and crop quality.”
On the other hand, we acknowledge the Reviewer’s concern since the extensive use of the term biostimulant may be not a straightforward definition and may lead to misunderstanding since it refers to a wide range of either biotic application or abiotic compounds.
We have modified the extensive use of biostimulant throughout the text as follow
- Abstract (line 30): “Targeted data evaluation for metabolites quantification further highlighted an eliciting effect of HS-ART and CT-COF on the synthesis of aromatic amino acids and phenolic compounds for nutraceutical application.”
- Introduction (line 72):” In this context, the natural organic fraction obtained from recycled biomasses, such as humic substances (HS) and compost teas (CTs) have been identified as valuable potential abiotic effectors or biostimulants due to their ability to influence directly and indirectly the plant metabolism”
- (Line 110): “…an associated goal is to strengthen a comprehensive understanding on the use of recycled biomasses as renewable source of tailored bioactive molecules for the biosynthesis of defined metabolites in aromatic plants.”
- (Line 117): “The cutting-edge challenge is the potential definition of viable protocol to correlate the characteristics of recycled natural organic components to yield and efficacy of specific plant products with pharmaceutical application”
- Discussion (Line 381): “The acknowledged stimulative actions of HS or CTs on plant bio-chemical and physiological activity have been related with…..”
- (Line 465): “The applied plant metabolomics workflow endowed the elucidation of main changes promoted by organic abiotic effectors in metabolic composition of plant extracts”
- (Lines 457-458): “Conversely, the average dissolved amount (50 mg L -1) of compost extracts may have undergone the above-sketched structural rearrangement with a suitable interaction with plant rhizosphere and unfolding of bioactive effects depending on the specific molecular components.”
Round 2
Reviewer 1 Report
After corrections, I consider the manuscript able for publication!
Reviewer 2 Report
Figure 6: The "Shikimic pathway" is still not completely represented, left part is missing. Please modify it.